# Single-block rockfall dynamics inferred from seismic signal analysis

Clément Hibert[1], Jean-Philippe Malet[1], Franck Bourrier[2], Floriane Provost[1], Frédéric Berger[2], Pierrick Bornemann[1], Pascal Tardif[2], and Eric Mermin[2]

[1]Institut de Physique du Globe de Strasbourg - CNRS UMR 7516, University of Strasbourg/EOST, 5 rue Descartes, 67084 Strasbourg, France
[2]Institut national de recherche en sciences et technologies pour l'environnement et l'agriculture (IRSTEA), 2 Rue de la Papeterie, 38402 Saint-Martin-d'Hères, France

*Correspondence to:* Clément Hibert (hibert@unistra.fr)

**Abstract.**

Seismic monitoring of mass movements can significantly help to mitigate the associated hazards, however the link between event dynamics and the seismic signals generated is not completely understood. To better understand these relationships, we conducted controlled releases of single blocks within a soft-rock (black marls) gully of the Rioux-Bourdoux torrent (French Alps). 28 blocks, with masses ranging from 76 kg to 472 kg, were used for the experiment. An instrumentation combining video cameras and seismometers was deployed along the traveled path. The video cameras allow reconstructing the trajectories of the blocks and estimating their velocities at the time of the different impacts with the slope. These data are compared to the recorded seismic signals. As the distance between the falling block and the seismic sensors at the time of each impact is known, we were able to determine the associated seismic signal amplitude corrected for propagation and attenuation effects. We compared the velocity, the potential energy lost, the kinetic energy and the momentum of the block at each impact to the true amplitude and the radiated seismic energy. Our results suggest that the amplitude of the seismic signal is correlated to the momentum of the block at the impact. We also found relationships between the potential energy lost, the kinetic energy and the seismic energy radiated by the impacts. Thanks to these relationships, we were able to retrieve the mass and the velocity before impact of each block directly from the seismic signal. Despite high uncertainties, the values found are close to the true values of the masses and the velocities of the blocks. These relationships allow to better understand the physical processes that control the source of high-frequency seismic signals generated by rockfalls.

## 1 Introduction

Understanding the dynamics of rockfalls and other mass movements is critical to mitigate the associated hazards but is very difficult because of the limited number of observations of natural events. With the densification of the global, regional and local seismometer networks, seismic detection of gravitational movements is now possible. The continuous recording ability of seismic networks allows a reconstruction of the gravitational activity at unprecedented time scale and the monitoring of unstable slopes (e.g. Amitrano et al., 2005; Helmstetter and Garambois, 2010; Levy et al., 2011; Burjánek et al., 2012; Panzera et al., 2012; Galea et al., 2014). More than the detection of these events, recent advances allow determining the dynamics

of the largest landslides on Earth from the very low-frequency seismic waves they generate. Inversion and modeling of the long-period seismic waves permits to infer the force imparted by these catastrophic events on Earth, and to deduce dynamic parameters (acceleration, velocity, trajectory) as well as their mass (Favreau et al., 2010; Schneider et al., 2010; Moretti et al., 2012; Ekström and Stark, 2013; Allstadt, 2013; Yamada et al., 2013; Hibert et al., 2014a, c). However, these approaches are limited by the size of the events. Only the largest landslides will generate the long-period seismic waves used in the inversion and the modeling methods. Moreover these events constitute only a small proportion of the landslides that occur worldwide.

In recent years, a new approach based on the analysis of the high-frequency seismic signal has been proposed. High-frequency seismic waves are generated independently of the size of the event, and can be recorded if seismometers are close enough to the source. Hence, this allows a seismic detection of the events that do not generate long-period seismic waves (e.g. Deparis et al., 2008; Helmstetter and Garambois, 2010; Dammeier et al., 2011, 2016; Hibert et al., 2011, 2014b; Clouard et al., 2013; Chen et al., 2013; Burtin et al., 2013; Tripolitsiotis et al., 2015; Zimmer and Sitar, 2015). The limitation of this approach is that high-frequency seismic waves are more prone to be influenced by propagation effects (attenuation, dispersion, scattering) and, more importantly, that the source of the high-frequency seismic waves associated with gravitational instabilities is not yet well understood.

Studies have shown that several landslide properties can be linked to features of the high-frequency seismic signals. In some cases, it has been observed that the landslide volume is correlated to the amplitude (Norris, 1994; Dammeier et al., 2011) or to the radiated seismic energy of the high-frequency signals (Hibert et al., 2011; Yamada et al., 2012). Other studies have shown that the high-frequency seismic signals can also carry information on landslide dynamics. Schneider et al. (2010) have determined with numerical modeling that a good correlation exists between the short-period seismic-signal envelope, the modeled friction work rate and the momentum (product of the mass and the velocity) for two rock-ice avalanches. The model-based approach proposed by Levy et al. (2015) predicts that a correlation can be found between the modeled force and the power of the short-period seismic signal for rockfalls that occurred at the Soufrière Hills volcano on Montserrat Island. Hibert et al. (2017) have demonstrated that, for 11 large landslides that occurred worldwide, the bulk momentum controls at the first order the amplitude of the envelope of the generated seismic signals filtered between 3 Hz and 10 Hz. These authors also demonstrated that the maximum amplitude of the seismic signal, corrected for propagation effects, is quantitatively correlated with the bulk momentum. These results are important as they open the perspective to quantify landslide dynamics, independently of their size, and directly from the seismic signals they generate (i.e. without inversion or modeling). Being capable of quantifying landslide properties directly from the seismic signals they generate is critical for the development of future methods aimed at their real-time detection and characterization using high-frequency seismic signals. However, before considering an operational implementation of such methods, we need to better understand the source of the generated high-frequency radiation and its link with landslide dynamics

One of the assumptions that emerge from these studies to explain the link between the landslide dynamics and the high-frequency seismic signal features is that this relationship can potentially originate from small-scale processes within the landslide mass, and between the landslide mass and the substrate. The dynamic properties of a bouncing particle within a granular flow might control the impulse imparted to the solid Earth at each impact, and the amplitude of the seismic wave generated

might be proportional to the magnitude of the impulse. However, this assumption raises an important issue: what is the link between the dynamics of a single bouncing particle (a rock for example) and the seismic signal generated?

Theoretical developments, laboratory and field experiments were conducted by Farin et al. (2015) to address this issue. These authors have shown that the mass and the speed of an impactor can be related to the radiated elastic energy and to the spectrum of the signal, following analytic developments based on the Hertz theory of impact (Hertz, 1882). However, the field experiment conducted showed that, in this case, these simple relationships did not perform well to quantify the velocity and the mass of single rocks from the seismic signal it generates. Difficulties to synchronize the seismic signals with direct observations and the use of a seismometer that was not capable to record the high-frequency energy of the generated seismic waves might explain why the analytic relationships were not confirmed by this experiment.

In this study we propose a new field experiment of controlled releases of single blocks to investigate the relationships between block properties and dynamics, and the features of the seismic signals generated by impacts with the slope. We deployed several short-period and broadband seismic stations to record the high-frequency seismic signal generated at each impact. The trajectory of each block is reconstructed with video cameras that were synchronized with the seismometers. The seismic signal processing allowed us inferring the amplitude of the seismic signal at the source, corrected for propagation effects, and the seismic energy radiated by the impacts. We then compare the features of the seismic signal of each impact to the dynamics and the properties of the released block.

## 2   The Rioux-Bourdoux experiment

The Rioux-Bourdoux controlled releases experiment focus was to study the seismic signal of single-block rockfalls on un-consolidated soft-rock, which is highly attenuating for seismic waves. The Rioux-Bourdoux is a torrent located in the French Alps, approximately 4 km north of the town of Barcelonnette (France). The slopes surrounding the torrent consist of Callovo-Oxfordian black-marls and are representative of the slope morphology of marly facies observed in south-east France. Due to the high erosion susceptibility of black marls numerous steep gullies have formed on these slopes.

We conducted the releases within one of these gullies (Figure 1a and b). The advantage of launching the blocks in a gully is that for every block the traveled path is roughly the same. Moreover, the steepness of the gullies that developed in black-marls allows the block to rapidly reach a high velocity. The travel path had a length of approximately 200 m and slope angles ranging from $\sim 45$ degrees on the upper part of the slope to $\sim 20$ degrees on the terminal debris cone. 28 blocks with masses ranging from 76 to 472 kg were manually launched.

Two video cameras (Sony alpha7 - 25 frames per second) were deployed at the base of the gully, close to the torrent. Ground-control points were marked for visual recognition on the videos and their 3D coordinates were measured by Global Navigation Satellite System (GNSS). A reference Digital Elevation Model (DEM) at a spatial resolution of 0.5 m was built from terrestrial LIDAR acquisitions (Figure 1c). The time of the cameras was set to be synchronous with the seismic sensors time (GPS). The seismic network was composed of 1 broadband seismometer (CMG40T - sampling frequency 100 Hz) located north of

the gully, and an antenna of 4 short-period seismometers (one 3 component and three with 1 vertical component - sampling frequency 1000 Hz) located south of the gully (Figure 1c).

## 3 Methods

### 3.1 Trajectory reconstruction and dynamic parameters estimation

To reconstruct the trajectory, the impacts of each block were manually picked on the frames of the videos. Thanks to the control points, the frames of the videos were projected on the DEM. Hence, once an impact was identified on the frame, the position of the pixel was reported on the DEM, which gave the true position of the impact. This processing was repeated for the two cameras, which gave an estimate of the uncertainties on the determination of the position and the time of the impact. The velocity just before impact was derived from the block trajectory and the duration of block flight before impact. The kinetic

energy was computed as:

$$E_k = \frac{1}{2}mV^2, \tag{1}$$

with $m$ the mass of the block and $V$ the velocity before impact. We also determined the potential energy lost during the block flight before impact from the difference of altitude of the block between two impacts, inferred from the reconstructed trajectory, as:

$$E_p = mg(h_{t_1} - h_{t_2}), \tag{2}$$

with $g$ the gravitational acceleration, and $h_{t_1}$ and $h_{t_2}$ the altitudes of the block at the impacts that occurred at the two successive times $t_1$ and $t_2$. Unfortunately, the resolution of the cameras and the complex dynamics of the blocks during the first seconds of propagation did not allow us to identify clearly the impacts on the upper part of the slope. However the trajectories of the blocks on the lower part of the slope were well constrained, with an average uncertainty on the inferred

velocity of the blocks before impacts of $0.95\,\mathrm{m\,s^{-1}}$, for velocities with values comprised between $6\,\mathrm{m\,s^{-1}}$ and $17\,\mathrm{m\,s^{-1}}$.

### 3.2 Seismic signal processing

Several authors have shown that the seismic waves generated by gravitational instabilities are dominated by surface waves (e.g. Deparis et al., 2008; Hibert et al., 2011; Dammeier et al., 2011; Levy et al., 2015). These high-frequency seismic surface waves are subjected to strong propagation effects, especially in a highly attenuating medium such as black marls. Figure 2 shows the

25 seismic signals recorded for the launch of the block number 4. The attenuation is visible when comparing peaks in the seismic signal recorded at the station located on the upper part of the slope (Figure 2a) to the ones recorded at the station on the lower part of the slope (Figure 2c), for the same time. The amplitude of the peaks is clearly dependent on the distance between the

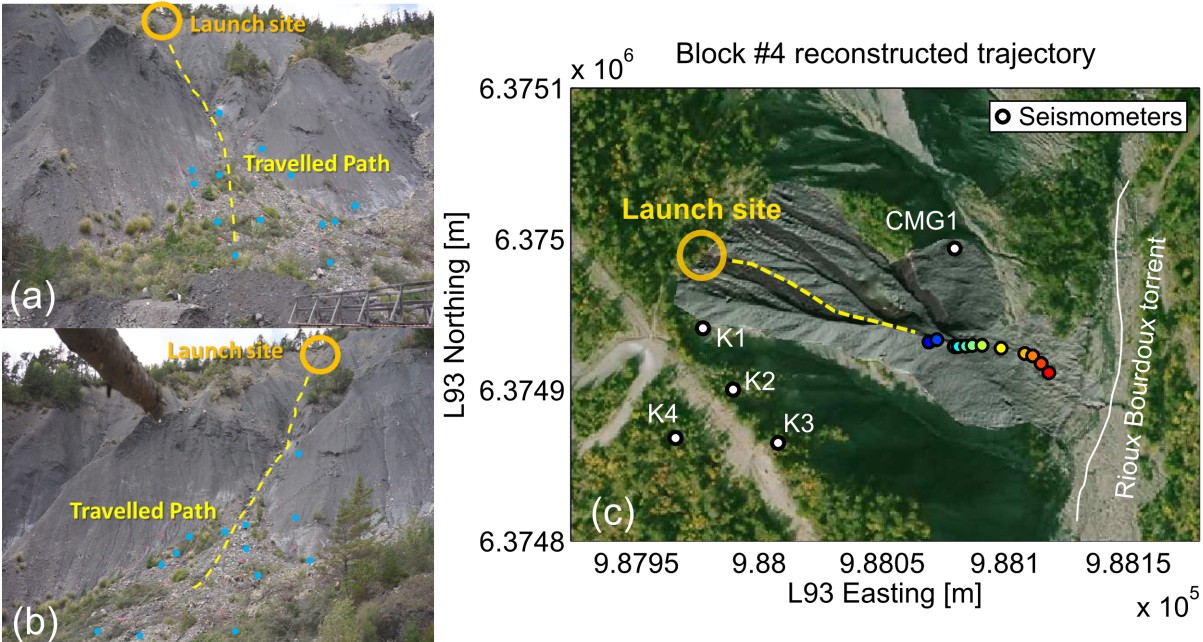

**Figure 1.** View from a) the first and b) the second video cameras deployed at the bottom of the slope. The ground control points are indicated by blue points. c) Trajectory reconstruction for block 4 on the DEM, built from LIDAR acquisition, superimposed on an orthophoto of the Rioux-Bourdoux slopes. Each point indicates the position of an impact and the color gradient represents the chronology of these impacts (blue for the first impact and red for the last one). K2 is a three-component short-period seismometer and K1, K3 and K3 are vertical-only seismometers. CMG1 is a broad-band seismometer.

impact and the seismic station. Moreover, Figure 2b shows the attenuation of the highest frequency with the distance of the source to the seismic station. To compare seismic signal features to the dynamic parameter of the rockfall, we have to correct these attenuation effects. Aki and Chouet (1975) proposed a simple attenuation law giving the amplitude $A(r)$ of a seismic surface wave recorded at a distance $r$ as:

$$5 \quad A(r) = \frac{1}{\sqrt{r}} A0 \times e^{-Br}. \tag{3}$$

If the distance between the station and the source is known, the computation of the amplitude at the source $A0$ is straightforward. However we have to determine the frequency dependent parameter $B$ that accounts for the anelastic attenuation of seismic waves. If we consider $r_i$ the distance between the source and station $i$ and $r_j$ the distance to station $j$, the apparent anelastic attenuation parameter $B_{ij}$ is then:

$$10 \quad B_{ij} = \frac{log(A(r_i)\sqrt{r_i}) - log(A(r_j)\sqrt{r_j})}{\sqrt{r_j} - \sqrt{r_i}}. \tag{4}$$

By combining Eq. (3) and Eq. (4), we can compute the amplitude at the source $A0$ for each pair of stations. This value is then averaged over all the pairs of stations, and the standard deviation gives an estimate of the uncertainty.

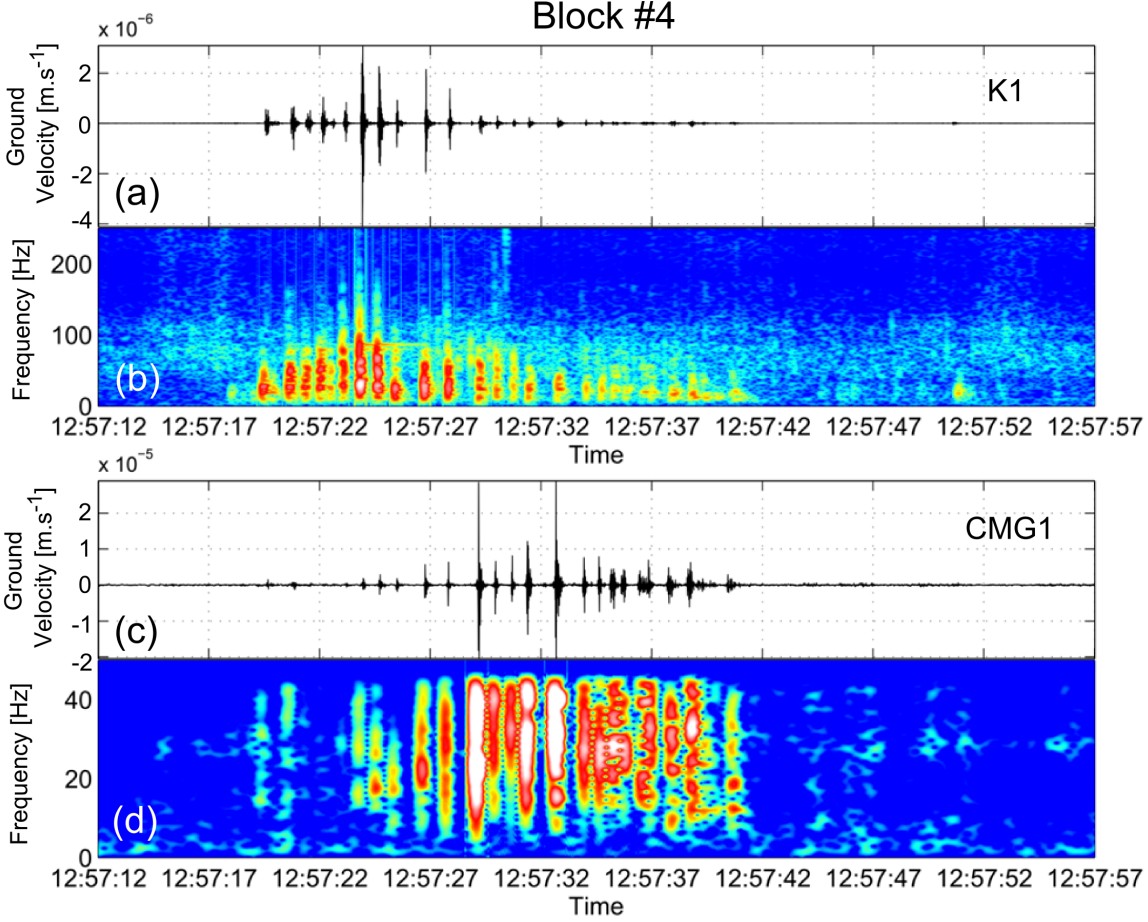

**Figure 2.** a) Signal recorded at the short-period station located on the upper part of the slope and b) corresponding spectrogram, generated by block number 4 (mass of 209 kg). c) Signal recorded at the broadband station located on the lower part of the slope and d) corresponding spectrogram, generated by block 4.

Another quantity that we want to compare to the dynamics of the block is the radiated seismic energy. The energy of a seismic surface wave can be computed as (Crampin, 1965):

$$5 \quad E_s = \int_{t_i}^{t_f} 2\pi r Dhc u_{env}(t)^2 e^{Br} dt, \tag{5}$$

with :

$$u_{env}(t) = \sqrt{u(t)^2 + Ht(u(t))^2},$$ (6)

where $Ht$ is the Hilbert transform of the seismic signal $u(t)$ used to compute the envelope $u_{env}(t)$, $t_i$ and $t_f$ the times of the beginning and the end of the seismic signal respectively, $h$ the thickness and $D$ the density of the layer through which the generated surface waves propagate, and $c$ their phase velocity. The average velocity of surface waves in black-marls formations observed in the area of the Rioux Bourdoux torrent is approximately $300\,\mathrm{m\,s^{-1}}$ (Hibert et al., 2012; Gance et al., 2012), which, for seismic signal with central frequencies around $f = 20$ Hz as observed on Figure 2, gives a propagation depth $h$, computed as $h = c/f$, of $\sim 15$ m. The density $D$ of dry black-marls is approximately $1450\,\mathrm{kg\,m^{-3}}$ (Maquaire et al., 2003).

Before computing the amplitude at the source and the energy of the seismic signals generated by impacts, we first selected the seismic signals with the following criteria. We excluded the seismic signals generated when i) sliding of the blocks occurred, ii) the blocks stopped mid-slope and iii) more generally when the signal-to-noise ratio was too weak on the seismic stations to perform the computation of the apparent anelastic attenuation parameter $B_{ij}$. $B_{ij}$ is dependent on the frequency of the seismic waves. Therefore the seismic signals were band-pass filtered between 1 and 50 Hz. This frequency band is chosen because most of the seismic wave energy is not attenuated in this band within the span of the seismic network (Figure 2b and d). For each seismic record selected, we manually picked the peaks corresponding to the impacts on each station. This processing results in a data set of 37 impact seismic signals, coming from 9 out of the 28 launches.

## 4  Results

### 4.1  Correlation between dynamic parameters and seismic signal features

From the reconstructed trajectories we inferred the velocity, the momentum and the kinetic energy of the block before each impact (Eq. 1), and the potential energy lost during the block trajectory before impact (Eq. 2). The velocities exhibit a low variability, with values ranging from $6\,\mathrm{m\,s^{-1}}$ to $17\,\mathrm{m\,s^{-1}}$ (Figure 3). We did not find significant correlation between the mass and the impact velocity.

The seismic signal processing yielded the maximum amplitude at the source $A0_{max}$ and the radiated seismic energy $E_s$ at each impact. The average uncertainty on the computation of the maximum amplitude $A0_{max}$, inferred from the standard deviation, and expressed as a percentage of the computed values (i.e. $A0_{max} \pm x\% A0_{max}$), ranges from 7% to 129%, and is 58% in average. Regarding the computation of the radiated seismic energy $E_s$, the uncertainty, estimated following the same approach, ranges from 55% to 152% of the computed values, and is 86% in average.

We investigated the possible correlations between: 1) the maximum amplitude at the source $A0_{max}$ of the seismic signal and the absolute momentum $|p|$ before the impact; 2) the radiated seismic energy $E_s$ and the potential energy lost $E_p$; 3) the radiated seismic energy $E_s$ and the kinetic energy $E_k$ before impact; and 4) the radiated seismic energy $E_s$ and the mass $m$ of the blocks. The analysis based on the Hertz's theory of impact conducted by Farin et al. (2015) yielded the parameter

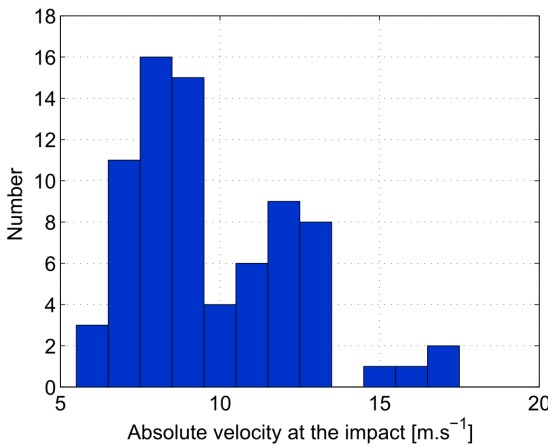

**Figure 3.** Histogram of the observed absolute velocities before impact.

$mV_z^{13/5}$, with $m$ the mass of the block and $V_z$ the vertical velocity before impact, that should in theory scale with the radiated seismic energy $E_s$ of the seismic signal generated at each impact. However, when investigating this relationship for real single-block rockfalls, they did not found a significant correlation with this parameter. The best correlation they found was with the parameter $mV_z^{0.5}$. We also investigated these two cases with our data set. We computed for each pair of parameters the

Spearman rank correlation coefficient $\rho$ and the corresponding $p-values$ (Table 1) (Spearman, 1904) as we assume that the parameters should scale following monotonic laws.

The best correlation coefficient $\rho$ has a value of 0.70 for the pair of parameters $E_s$ and $E_k$. Slightly lower correlation coefficient values are observed between the maximum amplitude $A0_{max}$ and the absolute momentum $|p|$ ($\rho = 0.67$) and the radiated seismic energy $E_s$ and the potential energy $E_p$ ($\rho = 0.68$). The correlation coefficient between the radiated seismic

energy $E_s$ and the best empiric parameter $mV_z^{0.5}$ found by Farin et al. (2015) is poorer ($\rho = 0.62$) than the one observed between the radiated seismic energy and the parameter $mV_z^{13/5}$ they derived from the Hertz theory of impact ($\rho = 0.69$). Finally, our results show that there is no correlation between the maximum amplitude $A0_{max}$ and the radiated seismic energy $E_s$ ($\rho = 0.44$) and between the radiated seismic energy $E_s$ and the mass of the blocks $m$ ($\rho = 0.51$). We also investigated other correlations between dynamic parameters and seismic signal features, with the vertical momentum or the vertical kinetic

energy for example, but we were unable to improve on the correlations found with the modulus of the dynamic quantities.

To characterize the relationships between the parameters that are correlated, we computed the regression lines that best fit the data (Figure 4 and Table 1). According to the theoretical analysis conducted by Farin et al. (2015), the dynamic parameters should scale proportionally with the seismic features. However several studies have shown that linear relationships allow a better fitting of the data gathered from the observation of natural events (e.g. Deparis et al., 2008; Dammeier et al., 2011;

Hibert et al., 2011). We computed the regression coefficients of the best fitting lines for the two types of relationships and assessed the quality of the fitting by computing the coefficients of determination $R^2$.

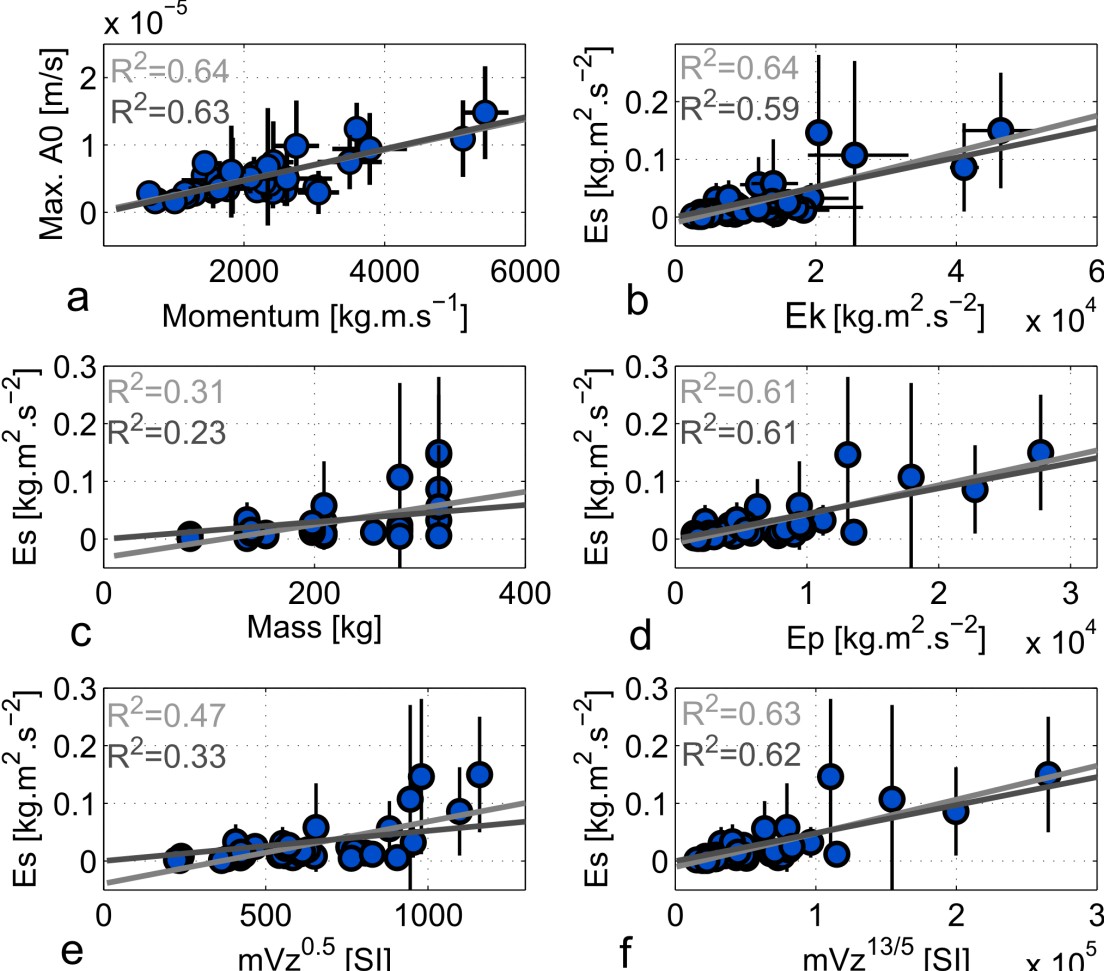

**Figure 4.** a) Maximum of the amplitude $A0_{max}$, corrected from attenuation, as a function of the average momentum $|p|$ of the block before the impact. Radiated seismic energy $E_s$ of the seismic signal generated at the impact as a function of: b) the kinetic energy before the impact $E_k$; c) the masses $m$ of the blocks; d) the potential energy lost $E_p$; e) the parameter $mV_z^{0.5}$; f) the parameter $mV_z^{13/5}$. Errors bars resulting from the computation of the momentum, the kinetic energy and the amplitude at the source are indicated by black lines. For each pair of parameters the light-gray line corresponds to the best regression line computed for a linear relationship and the dark-gray one to the best regression line computed for a proportional relationship.

Overall the $R^2$ coefficient values do not exceed 0.64 (Table 1). This is caused by a high scattering of the data which comes from the high uncertainties on the computation of the seismic attenuation parameters and hence on the values of $A0_{max}$ and $E_s$, as shown by the large error bars on Figure 4. The best $R^2$ coefficients are yielded by the linear regression between the maximum amplitude $A0_{max}$ and the momentum $|p|$, and the radiated seismic energy $E_s$ and the kinetic energy $E_k$ ($R^2 = 0.64$ for both

cases). For the couple of parameters $E_s/E_p$ and $E_s/mV_z^{13/5}$, $R^2$ coefficients are slightly lower, with values of 0.61 and 0.63 respectively. The regression of each pair of parameters by proportional relationships gives lower values for the coefficient $R^2$. However the $\beta$ coefficients of the best linear regressions are close to 0. We assume that linear regressions allow to better accommodate for the scattering of the data than proportional regressions, and that $\beta$ coefficients are not physically significant.

## 4.2 Retrieving block properties and dynamics from the seismic signal

We have shown that correlations exist between several dynamic quantities and features of the seismic signal generated at each impact. In this section we investigate if these relationships can provide accurate estimates of the mass and the velocity of the blocks, directly from the features of the seismic signals generated by the impacts.

Our results show that the maximum amplitude and the seismic energy are not correlated (Table 1). Hence we can combine the linear relationships inferred for the maximum amplitude and the momentum, and for the radiated seismic energy and the kinetic energy, with the coefficients $\alpha$ and $\beta$ yielded by the linear regressions. We can express the mass $m_i$ as a function of $A0_{max}$ and $E_s$ as:

$$m_i = \frac{5.9 \times 10^{11}(A0_{max} - 2.50 \times 10^{-7})^2}{(Es + 0.01)}. \tag{7}$$

Using Eq. (7), we computed $m_i$ for each impact of each block for which we were able to compute $A0_{max}$ and $E_s$, and compared the average estimates of $m_i$ to the measured mass $m_r$ of each block (Table 2). Overall, the inferred masses $m_i$ are close to the real masses $m_r$ of the block. However, the uncertainty on the inferred values is high, especially for blocks for which we have a few number of exploitable impacts and therefore few estimates of $A0_{max}$ and $E_s$. This may also come from the uncertainties related to the computation of the seismic quantities.

We can also estimate the velocity of the block before each impact using the linear regression and the corresponding coefficients found between the maximum amplitude $A0_{max}$ and the maximum momentum $p$, or between the seismic energy $E_s$ and the kinetic energy $E_k$, and with the masses inferred with Eq. 7. We choose to use the linear relationship between the amplitude and the momentum because the uncertainties associated with determining the amplitude at the source are lower than those associated with the radiated seismic energy. The inferred velocity $V_i$ can be computed as:

$$V_i = \frac{A0_{max} - 2.50 \times 10^{-7}}{2.26 \times 10^{-9} m_i}. \tag{8}$$

Figure 5a shows the distribution of the absolute difference between the velocities inferred $V_i$ and the velocities $V_r$ derived from the trajectory reconstruction. The values of the difference are comprised between 0.1 and 13.7 $\mathrm{m\,s}^{-1}$, with a median value of 2.4 $\mathrm{m\,s}^{-1}$. We also computed the ratio of the velocity absolute $|V_i - V_r|$ difference over the velocity derived from the trajectory reconstruction $V_r$ (Figure 5b). The majority of the values of the ratio falls below 0.5 (i.e. the difference is less than 50% of the value of the velocity derived from the trajectory reconstruction), and the median ratio is 0.2 (i.e. 20% of the value of the velocity derived from the trajectory reconstruction).

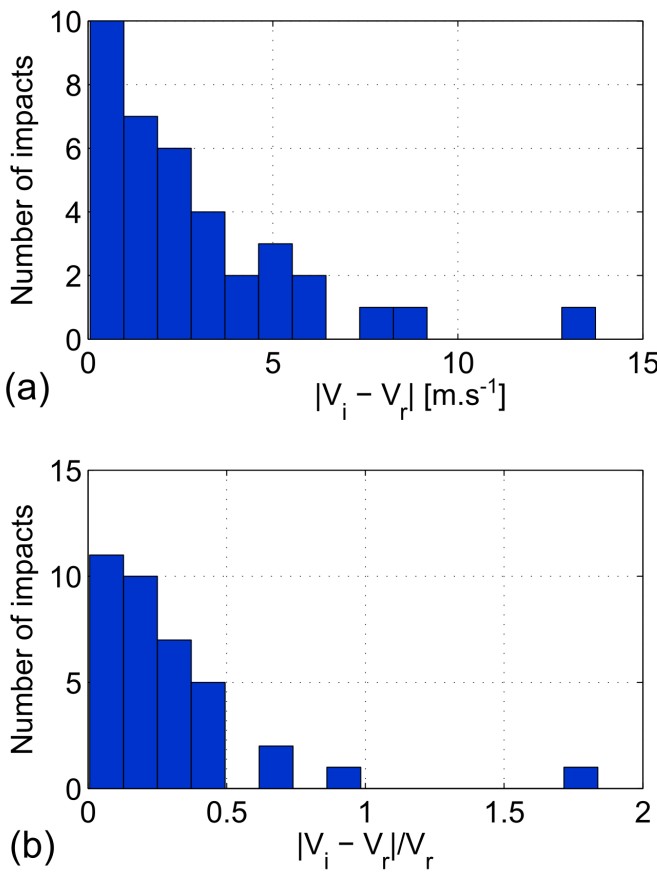

**Figure 5.** a) Histrogram showing the distribution of the difference between the velocity before impact $V_i$ inferred using Eq. 7 and Eq. 8 and the velocity $V_r$ estimated from the video cameras; b) Same as a) but normalized by the value of the velocity $V_r$ estimated via the video cameras.

## 5 Discussion and conclusion

The Rioux-Bourdoux experiment of controlled single-block rockfalls produced important results to better understand the links between the dynamics of rockfalls and the seismic signal associated. Our results suggest that correlations exist between the seismic signal features and the energy, the velocity and the mass of single-block rockfalls. We observed that the maximum amplitude of the seismic signal generated at each impact and the momentum (product of the mass and the velocity) of the blocks are correlated. Our results also suggest that the energy of the seismic radiation released at each impact scales linearly with the potential energy lost and the kinetic energy.

By combining the scaling laws found, we were able to infer realistic values of the masses and the velocities before impact of the blocks from the amplitude and the energy of the seismic signals generated at each impact. The difference between the mass of the blocks determined from the seismic quantities and the real values is 27% in average. Our results also demonstrate that when the number of impact seismic signals used to determine the mass of the blocks increases, the error made on the inferred values decreases. For the velocities, the average difference between the inferred and the real values of the velocity is 20%. These errors might come from the uncertainties on the computation of the seismic quantities. We determined, from the computation of the seismic quantities on multiple pairs of stations, that the average uncertainties are 58% and 86% on the computed values of the amplitude at the source and of the radiated seismic energy respectively. We suppose that these uncertainties are mainly caused by the simple seismic attenuation model used.

We found that the relationship derived from the Hertz's theory of impact proposed by Farin et al. (2015) that links the radiated seismic energy of the signal generated to the parameter $mV_z^{13/5}$ is verified with our data. However the scaling between the seismic energy and the parameter $mV_z^{13/5}$ did not yield significantly better quantitative correlation than the one observed between the radiated seismic energy and the kinetic energy, or between the amplitude at the source and the momentum of the block before impact ($\rho = 0.69, 0.70$ and $0.67$ respectively). This confirms the combined role of the mass and the velocity before impacts of the block in the generation of seismic waves, but does not allow us to identify a unique dynamic parameter that would control the seismic signal features. Further analytical and theoretical developments are needed to understand the physical processes that explain these correlations, and ultimately what are the physical parameters that control the characteristics of the seismic signal generated.

An issue that arose from studies on the link between the seismic signals and the dynamics of mass movements is about the energy transfer and more specifically the ratio $R_{s/p}$ between the radiated seismic energy and the potential energy lost. Deparis et al. (2008) found for 10 rockfalls that occurred in the French Alps that this $R_{s/p}$ ratio is comprised between $10^{-5}$ and $10^{-4}$. Vilajosana et al. (2008) have found a $R_{s/p}$ ratio of $10^{-3}$ for an artificially triggered rockfalls in the Montserrat massif (Spain). In volcanic contexts, Hibert et al. (2011) and Levy et al. (2015) have observed $R_{s/p}$ ratios ranging from $10^{-5}$ to $10^{-3}$. In this study, we found a $R_{s/p}$ ratio between the radiated seismic energy and the potential energy lost of approximately $10^{-6}$ (Table 1). Interestingly a ratio of the same order is observed between the radiated seismic energy and the kinetic energy. The value of the $R_{s/p}$ ratio is lower than those observed in other contexts. We assume that this might be explained by the nature of the substrate as in our case the rockfalls propagated on soft rocks, which may absorb more potential energy (by deformation for example) than igneous (Hibert et al., 2011; Levy et al., 2015) or metamorphic hard rocks (Deparis et al., 2008; Vilajosana et al., 2008). Investigating this assumption on the role of the substrate on energy transfer by replicating the experiment of controlled releases of single blocks in other contexts constitutes one of the perspectives of this work.

We identified several limitations that have to be addressed before considering an operational application of seismology to quantify rockfall properties. First, our results show that better attenuation models are needed to reduce the uncertainties on the computation of the seismic signal features. This could be achieved by deploying denser seismic networks for example. Second, the range of the mass of the blocks used in our experiment spans only one order of magnitude. The behavior of the relationships we found has to be investigated for a larger range of volumes. Third, the relationships found may be specific to

a particular context and may depend on the substrate onto which the rockfalls propagate. This again underlines the relevance and the necessity of reproducing similar studies in new contexts.

Finally, our results give a new insight on the processes that generate high-frequency seismic signals associated with rockfalls, landslides, rock-avalanches, and granular flows in general. We show that the maximum amplitude of the seismic signal generated by the impact of a single particle is proportional to its mass and velocity. In a granular flow, a very large quantity of particles interact with themselves and with the substrate at a given time. The magnitude of these impulses imparted on the Earth by each particle might be controlled by the mass and the velocity of the particles within the flow according to the correlations we observed. The issue is now to understand what controls the dynamics of the particles within the flow and how their complex interactions influence the generation of seismic waves. This should be more thoroughly investigated, using numerical granular flow models for example, and is probably the key to model the high-frequency seismic signal associated with gravitational instabilities in the future.

## 6 Code and Data availability

The codes and the data used in this study are accessible upon request by contacting C. Hibert (hibert@unistra.fr).

*Author contributions.* C. Hibert, J.-P. Malet, F. Bourrier and F. Provost participated in the acquisition and the processing of the seismic and kinematic data. F. Berger, P. Tardif and E. Mermin helped to design and perform the Rioux Bourdoux experiment, and for the acquisition of the video and the reconstruction of the trajectories of the blocks. P. Bornemann performed the Lidar survey and the processing of the data that allow reconstructing the DEM of the gully into which blocks have been launched.

*Competing interests.* The authors declare that they have no conflict of interest.

*Acknowledgements.* We are very grateful toward Anne Mangeney for helpful discussions and insightful suggestions, and to Georges Guiter (RTM) for organizing the practical details and the security of the experimental launch site. This work was carried with the support of the French National Research Agency (ANR) through the projects HYDROSLIDE 'Hydrogeophysical Monitoring of Clayey Landslides' and SAMCO 'Adaptation de la Société aux Risques Gravitaires en Montagne dans un Contexte de Changement Global' and of the Open Partial Agreement Major Hazards of Council of Europe through the project 'Development of cost-effective ground-based and remote monitoring systems for detecting landslide initiation'. The data were acquired using instruments belonging to the French national pool of portable seismic stations RESIF-SISMOB (CNRS-INSU). The authors gratefully acknowledge Maxime Farin and Francesco Panzera for insightful reviews. We also would like to thank the Editor, Jens M. Turowski, for his substantial comments, which helped to improve this manuscript.

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

| Parameters $(X,Y)$ | Spearman correlation | | Proportional | | | linear | | |
|---|---|---|---|---|---|---|---|---|
| | $\rho$ | $p-values$ | $\alpha$ | $\beta$ | $R^2$ | $\alpha$ | $\beta$ | $R^2$ |
| $A0_{max} = \alpha\,|p| + \beta$ | 0.67 | $1.12\,10^{-7}$ | $2.35\,10^{-9}$ | 0 | 0.63 | $2.26\,10^{-9}$ | $2.50\,10^{-7}$ | 0.64 |
| $E_s = \alpha\,E_p + \beta$ | 0.68 | $6.75\,10^{-6}$ | $4.40\,10^{-6}$ | 0 | 0.61 | $5.04\,10^{-6}$ | -0.01 | 0.61 |
| $E_s = \alpha\,E_k + \beta$ | 0.70 | $3.01\,10^{-6}$ | $2.59\,10^{-6}$ | 0 | 0.59 | $3.09\,10^{-6}$ | -0.01 | 0.64 |
| $E_s = \alpha\,m + \beta$ | 0.51 | $1.3\,10^{-3}$ | $1.48\,10^{-4}$ | 0 | 0.23 | $2.85\,10^{-4}$ | -0.03 | 0.31 |
| $E_s = \alpha\,mV_z^{13/5} + \beta$ | 0.69 | $4.16\,10^{-6}$ | $4.86\,10^{-7}$ | 0 | 0.62 | $5.85\,10^{-7}$ | -0.01 | 0.63 |
| $E_s = \alpha\,mV_z^{0.5} + \beta$ | 0.62 | $7.63\,10^{-5}$ | $5.24\,10^{-5}$ | 0 | 0.33 | $1.07\,10^{-4}$ | -0.04 | 0.47 |
| $A0_{max} = \alpha\,E_s + \beta$ | 0.44 | $8.2\,10^{-3}$ | - | - | - | - | - | - |

**Table 1.** Spearman correlation coefficients, coefficients of the regression lines for proportional and linear relationships and corresponding coefficient of determination $R^2$.

| Block # | $m_r$ [kg] | $m_i$ [kg] | std. | Nbr. impacts |
|---|---|---|---|---|
| 9 | 281 | 198 | 56 | 5 |
| 1 | 318 | 334 | 71 | 6 |
| 4 | 209 | 208 | 115 | 7 |
| 35 | 82 | 84 | 68 | 3 |
| 33 | 256 | 97 | - | 1 |
| 22 | 154 | 171 | 146 | 3 |
| 20 | 198 | 211 | 39 | 6 |
| 17 | 136 | 181 | 118 | 4 |
| 13 | 140 | 270 | 162 | 2 |

**Table 2.** Comparison between the real mass $m_r$ of the blocks and the average inferred masses $m_i$ computed with Eq. 7.