# Peer review of "Single-block rockfall dynamics inferred from seismic signal analysis"

_Earth Surface Dynamics, 2016_

## Referee Comment (RC1) · M. FARIN (Referee) · 13 Jan 2017

(Please read the supplementary PDF version of the review because the format of the online version is not good.)

The purpose of this paper is to investigate the link between the dynamics of a single block falling down a slope and the characteristics of the seismic signal radiated during the impacts of this block on the ground. The authors conducted a release of 28 blocks of masses m ranging from 76 kg to 472 kg on a steep slope constituted of unconsolidated soft-rock. Using a couple of cameras, they were able to retrieve the trajectories of the blocks and compute their speed of impact Vz, for each impact. In addition, the authors installed a network of 5 seismic stations, with sampling frequency 100 Hz for one station and 1000 Hz for the others, in order to record the seismic signals emitted

by the impacts of the blocks on the slope.

For a selection of rock falls for which the signal-to-noise ratio is high and for which the block did not roll and stop on the slope, the authors then computed the absolute seismic amplitude at the position of the impact and the absolute radiated seismic energy for each impact. The estimation of the absolute seismic amplitude and radiated energy required the authors to evaluate the attenuation of seismic amplitude with distance from the source and as a function of frequency.

For each of the selected impacts, the authors then compare (1) the absolute seismic amplitude with the momentum $p = mV_z$ of the block before the impact and (2) the radiated seismic energy $E_s$ with (i) the kinetic energy $E_k$ of the block before the impact, (ii) the potential energy $E_p$ lost by the block during the impact and (iii) a function of the mass m and speed of impact $V_z$ of the block derived analytically from Hertz model of elastic impact by Farin et al. (2015): $m V_z^{(13/5)}$. They also compare the radiated seismic energy $E_s$ with the parameter $mV_z^{(0.5)}$, that was observed to better fit the data for similar experiments of blocks impacts conducted by Farin et al. (2015). The authors chose to relate the seismic parameter X to the dynamic parameter Y using a linear relationship $X = aY + b$. The coefficient of regression $R^2$ varies from 0.31 to 0.64 with the best fits observed between the absolute seismic amplitude and the momentum $mV_z$ and between the radiated seismic energy $E_s$ and the kinetic energy $E_k$ of the block. Finally, they invert these empirical scaling laws in order to express the mass m and the speed $V_z$ of the blocks as functions of the seismic amplitude and the radiated seismic energy $E_s$. They use this scaling laws to retrieve the masses and speeds of the blocks that they compare with the real values. A relatively good correlation is observed, within $\pm$ 50% of the real value.

The authors conclude that their study show that there is a good correlation between the seismic amplitude and the momentum of the block before impact and that this may help to get information on granular flows dynamics from the generated seismic signal.

The paper is globally clear to read and the successive sections follow naturally each others. I find personally that it is interesting to have new data of seismic signals generated by block impacts and be able to evaluate the dynamics of the block in parallel in order to better understand the link between the two on the field. The authors took care to evaluate the dynamics of the block with a good precision, with an uncertainty less than 1 m sˆ-1 for block speeds varying from 6 m sˆ-1 to 17 m sˆ-1. When we compare seismic parameters to dynamic parameters, it is important to evaluate the absolute seismic parameters at the source because they strongly depend on the distance between the source and the instrument and on the frequency. Care has also been taken in evaluating absolute seismic parameters in this paper. Therefore I think the presented data are of good quality. However, I think that the paper needs a major revision before being considered for publication because it contains major confusions and misinterpretations of the data.

My main concern in is the fact that the authors say several times in the paper that they show a scaling (or proportionality) between the seismic amplitude and the momentum of the block while they are showing a linear relationship. There is a important confusion here because a scaling (or proportionality) is a relation Y = a X while a linear relationship (as showed in this paper) is Y = aX+b, with b a nonzero constant. This has a different implication for the interpretation of the data. The paper should be rewritten with this point in mind. This confusion is particularly problematic when the authors are comparing the parameter mVzˆ(13/5) derived by Farin et al. (2015) to the radiated seismic energy Es. They are testing a law Es = a mVzˆ(13/5) + b and claim that the fit of this law with their data is better than it was in the paper of Farin et al. (2015). However, the analytical scaling law established in Farin et al. (2015) and tested with their rockfall experiments was Es = a mVzˆ(13/5) (with b=0): this is a different law. In the present paper, the parameter b is not 0 and it is several orders of magnitude larger than the parameter a. The fit Es = a mVzˆ(13/5) (with b=0) should be tested instead. Moreover, since the parameter b does not exist in the analytical model, I do not know if this parameter has a physical meaning, even though it has the dimension of an energy.

Also, an analytical expression of the proportionality coefficient a is given in Farin et al. (2015). The exact law and empirical law (with the exact and empirical value of a) could be compared to the seismic energy Es. An interesting question when we study the seismic signal generated by rockfall is to establish their energy budget, i.e. determine the amount of kinetic energy or potential energy lost that is radiated in the form of elastic waves. In other words, I think the authors should compute the value of the ratios Es/Ek and Es/Ep (or maybe also Es/(Ek+Ep)). These ratios should be less than 1 and the rest of the kinetic and/or potential energy lost is dissipated in plastic deformation (irreversible deformation) of the ground or in viscoelastic processes (heat). These ratios can then be compared with that computed for larger rockfalls in the crater of the Piton de la Fournaise, La Reunion Island (Hibert et al. 2012) or with that obtained in other studies (e.g. Deparis et al. 2008). Thus we could see if the energy budget for one single impactor is different than for a rockfall constituted of several blocks. These ratios are proportionality relations between seismic and dynamic parameters. In a nutshell, I think that proportionality relationships Y=aX between seismic and dynamic parameters would have much more interesting implications for interpretations of the seismic signals generated by rockfalls than linear relationships Y=aX+b. Besides, no confusion should be made between the two kinds of relationship. A linear relationship may better fit the data of this paper than a proportionality law X = a Y but in this case, both fits (X = aY+b and X = aY) should be shown and a physical interpretation of parameter b should be given.

An other problem I see is when the authors want to retrieve the mass and the speed of the blocks from the seismic signal. Two seismic variables are used: the absolute seismic amplitude and the radiated seismic energy. However, I do not think these two variables are independent of each others. I would not be surprised if the radiated seismic energy is proportional to the squared absolute amplitude. In this case, the mass and the speed could be expressed as functions of the radiated seismic energy alone. The problem is that I don't think it is possible to retrieve two independent dynamic parameters from only one seismic variable. An advantage of the present study compared

with the previous ones (e.g. Farin et al. (2015)) is that the authors have access to higher frequencies up to 500 Hz, with respect to 50 Hz before. Therefore, they potentially have access to all the frequencies emitted during the impacts, contrary to the previous study. Thus an interesting seismic parameter to evaluate would be the mean frequency of the seismic signal. the analytical model of impact of Hertz shows that the mean frequency is inversely proportional to the mass m of the block. It would be interesting to test this scaling. The mean frequency of the signal is independent of the radiated seismic energy so if empirical scaling laws are established between these two parameters and the mass and the speed of the block, the laws can be inverted to retrieve the masses and the speeds. Farin et al. (2015) established two analytical scaling laws relating the mass and the speed of the block to the radiated seismic energy and the mean frequency of the signal, i.e. equations (29) and (30) of their paper. I would be curious to see if these equations can provide reasonable values of the masses m and the speeds Vz of the blocks with the present experiments. Maybe the absolute seismic amplitude and the radiated seismic energy are independent of each others. In that case it should be shown somewhere. Besides, if the mean frequency of the signal is not inversely proportional to the mass of the block, it would be interesting to show it. That would mean that Hertz's model does not apply on the field.

My following comments refer to specific lines in the paper.

The abstract needs a context sentence.

page 1 line 8, line 10. . ., 'Âăthe energy of the corresponding part of the seismic signalÂă' , 'the energy of the seismic radiationÂă',. . .Âătry to always call this energy is the same way all along the paper, for example 'Âăthe radiated seismic energyÂă' because it is sometimes difficult to understand to what energy you are referring to.

Âăpage 1 line 8: 'ÂăOur results suggest that the amplitude of the seismic signal scales with the momentum of the block at the impact.Âă'. No, be careful thorough in the paper: a scaling is a proportionality, not a linear relationship. This is important.

page 1, line 12: 'Âăthe masses and the velocitiesÂă' or 'Âăthe mass and the velocityÂă'

page 2, line 19: precise that this is true in the frequency range 3 Hz to 10 Hz

page 2, line 20: 'The authors also demonstrated that the maximum amplitude of the seismic signal, corrected from propagation effects, scales with the bulk momentumÂă': That was also a linear relationship, not a scaling (i.e. proportionality).

page 2, lines 27-32: this paragraph needs rewriting: line 30: 'ÂăThe impulse imparted to the solid Earth by a bouncing particle within a granular flow will be proportional to the kinematics of the particle, and the amplitude of the seismic wave will be proportional to the magnitude of the impulseÂă'. This sentence is not very clear, it particular 'proportional to the kinematics of the particleÂă' does not mean anythingÂă : I would rather say that the seismic amplitude is proportional to the impulse, which is itself proportional to the speed of the particle (in theory). Line 31: 'ÂăHowever, this assumption raises an important issue: what is the link between the dynamics of a single bouncing particle (a rock for example) and the seismic signal it generates?Âă'. Be more specific because you said just before that the seismic amplitude is proportional to the impulse.

page 2, l. 34: not exactly true: the mass and the speed of an impactor can be related to the radiated elastic energy and the mean frequency of the signal. Do not write 'Âăat a given frequencyÂă', it could be misinterpreted.

page 3, l.1: precise here what is the relation you are referring to: $E_s = a\, mv^{(13/5)}$.

page 3, l.4: It is very strange for me why you say that having frequency < 50 Hz is a limitation (which is true) but then you are filtering your signals below 50 Hz in the following (p 6, l 30). Why don't you take advantage of having frequencies higher than 50 Hz in order to improve the estimate of the masses and speeds of the blocks with respect to the previous study? Moreover, you know the mass and the speed of the blocks so you can evaluate a theoretical mean frequency of the signal generated by an impact using Hertz theory of impact and then compare with the measured mean frequency in your data. You would know if your seismic stations are sensitive to the whole frequency spectrum emitted during the impact or if you are loosing energy in the highest frequency (if the measured mean frequency is smaller than the theoretical mean frequency). This would help you to interpret the difference between the measured radiated seismic energy Es and the parameter a mVˆ(13/5): normally the measured Es should be smaller than a mVˆ(13/5) at high frequency if you are not sensitive to the highest frequencies. Finally, if you don't obtain any satisfying results using the mean frequency, it would mean that the mean frequency is not a reliable enough parameter to use to extract information from the seismic signal generated by impacts on the field: this is an interesting result.

page 3, l. 4, you should rather say 'Âǎa great part of the energy liberated at the impact is at high frequencies (> 50 Hz)Âǎ. . .'. An other important limitation we had was that there was no synchronization between the seismic signal and the movies. . .

page 3, l.6-11: You should better highlight what is new in your study with respect to the previous study: you use several seismic stations that can record higher frequencies, up to 500 Hz and a better identification of the seismic signals associated with each impact of the blocks.

page 3, l.25: Is the torrent producing a lot of seismic noise?

page 4, l.6: Define clearly what are the potential energy lost and the kinetic energy as a function of the mass and the speed of the block, and show these relations on the axis of Figure 4. Also: is the speed of impact Vz vertical or inclined with respect to the slope/vertical? Can you observe an effect of the angle of impact with respect to the normal to the slope on the radiated seismic energy? Are more inclined impacts less seismically efficient (lower Es/Ep) than more normal impacts? This might potentially explain part of the discrepancy.

page 4, l. 11: Write here the range of values the speed of impacts Vz can take because we don't know if 1 m sˆ-1 is a large and small uncertainty.

Figure 1c: not clear what the colored points are referring to. The text is the figure is too small, especially along the torrent.

page 5, l. 4, the Figure 2b also shows the attenuation of small frequencies. Rephrase the sentence.

page 5: It is not clear how you obtain the equation (2) and what the index ij are representing for B. You should directly say that B depends on frequency and show B as a function of the frequency, on a Figure for example (maybe in Appendix), so that we can know what is the quality factor Q. If you assume that B does not vary with frequency then give the value of B (or a range of values).

page 5, l. 23: Can you measure the wave speed in this specific site with your present seismic data by measuring the difference of time travel after an impact between several seismic stations?

page 6, l. 30: What a pity not to use the high frequencies > 50 Hz. There may a lot of interesting information in it.

Figure 4: Do you observe a correlation between radiated seismic energy Es and the squared momentum $|p|^2$ ? See my first comment about the law $X = a\,Y$ and the energy ratios. (d) Why is alpha negative ? The lost potential energy is not negative so it should not. See my first comment: the laws established and tested in Farin et al. (2015) are $Es = a\,mVz^{13/5}$ and $Es = a\,mVz^{0.5}$, not the ones you are showing. The caption of the figure can be simplified: 'ÂăDecimal logarithm of the seismic energy Es of the seismic signal generated at the impact as a function of (b) Ek, (c) Mass, (d) Ep, (e) $mVz^{0.5}$ and (f) $mVz^{13/5}$.Âă'

page 9, l. 3-9: Rewrite this paragraph in accordance with my first comment.

page 9, eq. (6): I am not sure that the absolute amplitude and the radiated elastic energy are independent variables. Also write explicitly the equation for the speed Vi as a function of the signal parameters.

Table 1, Fig 5: Represent the results of the inversion more uniformly: a figure with two plots showing (a) mi as a function of mr and (b) Vi as a function of Vr, with error bars and the line Y=X would be much clearer than a table and a histogram that mean to represent the same thing for m and Vz.

page 11, l.4: 'Âălinear scalingÂă' => linear relationships.

page 11, l. 7: 'Âăthe seismic radiation released at each impact scales linearly with the potential energy lostÂă': no.

page 11, paragraph 2: You did not verify this scaling law either. 'ÂăIn our study the instruments we deployed permitted to record most of the energy generated at impacts. This underlines the importance of choosing adequate seismometers, capable of recording the whole seismic energy generated at the impacts,Âă for future studies.Âă'Âă': Yes but you did not take advantage of this because you filtered the signals below 50 Hz while energy is clearly visible at more than 200 Hz on Figure 2a... Therefore you can not use this sentence to explain why you observed better correlations.

- page 11, last line: 'ÂăWe show that the maximum amplitude of the seismic signal generated by the impact of a single particle is proportional to its momentum.Âă' This is false.

- page 12, first line: 'The source of the seismic signal generated at this given time might therefore be the sum of the impulses imparted by the particles to the ground.Âă' I do not think we can say that because the signals emitted by two particles impacting the ground at roughly the same position and time can destruct or add themselves, depending of their phase. The energies of each impacts may be added, however (see the paper of Tsai et al. 2012 on the seismic noise of river: 'ÂăA physical model for seismic noise generation from sediment transport in riversÂă', GRL (2012).). Moreover, in the granular flow experiments we did with Anne Mangeney during my PhD (cf. Farin, M. (2015),ÂăÉtude expérimentale de la dynamique et de l'émission sismique des instabilités gravitaires. IPGP, France.), we showed that the scaling law that relate the radiated

seismic energy to the mass for a single impactor is not the same as for a granular flow of multiple particles of the same size. The relationship between the radiated seismic energy and the mass and the speed of the particles in a granular flows is much more complex than for one impact because all the particles are interacting with each others and each of them move in a random direction with respect to its neighbors (in an agitated flow) and each of them has a different fluctuating speed (instantaneous speed - mean speed of the flow). Therefore, the seismic amplitude generated by a granular flow does not simply scale (nor has a linear relationship) with the momentum of one particle in the flow. As you say in the last sentence, numerical models (DEM or statistical models like kinetic theories of granular gas) can help us better understand the complex link between particle/flow dynamics and seismic signal in granular flows.

Âă

Please also note the supplement to this comment:
http://www.earth-surf-dynam-discuss.net/esurf-2016-64/esurf-2016-64-RC1-supplement.pdf

---

## Referee Comment (RC2) · F. Panzera (Referee) · 16 Jan 2017

The manuscript "Single-block rockfall dynamics inferred from seismic signal analysis" by Hibert et al. is interesting and contains innovative information about seismic radiation due to rockfall. Below some comments that I hope help the authors in improving the manuscript.

Although I am not an English mother-tongue, at times I found some sentences difficult to follow. In my opinion, the authors should improve the English language.

In the Introduction, few lines should be added on the importance of rockfalls characterization, through seismic method (but not only). See for instance: Burjanek J., Moore J.R., Yugsi-Molina F.X., Fah D. (2012) Instrumental evidence of normal mode rock slope vibration. Geophys. J. Int., 188, 559–569. F. Panzera, S. D'Amico, A. Lotteri, P. Galea, G. Lombardo (2012) Seismic site response of unstable steep slope using noise measurements: the case study of Xemxija bay area, Malta. Nat. Hazard Earth Sci. Syst., 12, 3421–3431 doi: 10.5194/nhess-12-3421-2012 P. Galea, S. D'Amico, D. Farrugia (2014) Dynamic characteristics of an active coastal spreading area using ambient noise measurements – (Anchor Bay, Malta). Geophys. J. Int., 199, 1166–1175 doi: 10.1093/gji/ggu318

In Figure 1a and b, what do the authors indicate with blue points? Which is the meaning of coloured points in Figure 1c?

In Figure 1c, I understand that CMG1 is the broadband seismometer, but it is unclear which is the 3D short-period seismometer between K1, K2, K3 and K4. The authors should add a legend in map or some description in the figure caption.

The authors assume that seismic wavefield, generated by rockfalls, is composed mainly by surficial waves and consequently that the contribution of body waves is negligible. They must support this hypothesis through observations or by quoting references.

The authors assume that seismic wave velocity in black-marls is 300 m/s quoting as references Hibert et al. (2012) and Gance et al. (2012). Are the quoted studies performed in the same formations near Rioux Bourdoux? The sentence must be rewrite as follow: "The average velocity of surface waves in black-marls in the area of Rioux Bourdoux is approximately 300 m/s (Hibert et al., 2012; Gance et al., 2012)." or "The average velocity of surface waves in black-marls, considering information coming from literature, is approximately 300 m/s (Hibert et al., 2012; Gance et al., 2012)."

I suppose that the propagation depth is obtained by considering lampda=V/f. This assumption should by quoted in the text and the authors must specify why they chosen 20 Hz as central frequency for their computation.

The authors used a linear regression to interpolated their data. Did they try to use a

power or logarithm law using a lin-lin graph? The R2 for each linear regression curve should be visible in the graphs of Figures 4.

Probably in the case of x and y having uncertainties a Generalized Orthogonal Regression is need instead than standard least-squares.

The term "proportional" used in the manuscript is not correct, because the authors use a linear regression (y=ax+b) with a non-zero "b".

The authors should better highlight which are the application fields of their study and the novelty with respect to the previous studies.

---

## Editor Comment (EC1) · JM Turowski (Editor) · 21 Feb 2017

Dear authors,

We have now received two reviews, and, since the discussion closes in a few days, I give a short summary of what I find important. While both reviewers are generally supportive of the manuscript, they both raise some criticisms that need to be addressed before publication. I add some comments of my own reading to that.

Presentation and language Reviewer #2 criticizes presentation and language and asks for rewriting with a focus on clarity. I agree that there are fairly frequent odd formulations and unclear writing.

Scaling, linearity and the fits Reviewer #1 raises concerns about the terminology used

in the paper. I do not agree with his definitions; in my understanding, two variables A and B scale with each other, if they have a positive monotonic relationship, without the need of specifying a function. That is, if A increases, B increases also. Two variables are proportional if their ratio is a constant. And they are linearly related if A=mB+b, where m and b are constants. However, I agree with reviewer #1 that in the manuscript, the terminology is not used in a common way. For example, I would not say that two variables scale linearly, but rather that they have a linear relationship of linear dependence. That said, there is something funny about the plots in Fig. 4 and the way the relationships are discussed. The plots in Fig. 4 are all log-log. In this representation, a proportionality would result in a straight line with a gradient of one. A linear relationship would result in a curved line. A closer look reveals that the depicted fit lines actually have gradients that deviate from one. They show power law relationships. The fit values given in Table 2 indicate exponents of up to 2. This may change the entire results, discussion and outcomes of the paper. Here is at least a major problem with the communication, if not with the use of the fits and the statistical relationships. These need to be carefully resolved and communicated.

Relation to theory Reviewer #1 comments that the relations to his theory are partially incorrect and not well described. In light of the issue raised in the preceding point, I ask the authors to clearly present the used theory in the paper, identify appropriate hypotheses that can be tested with the data (both a theory-derived hypothesis and a null hypothesis), and to discuss how the outcomes of their experiments relate to these.

Energy budget I also like reviewer #1's suggestion of the energy budget and ask the authors to provide appropriate calculations and a discussion on this.

Relationship between seismic energy and amplitude The authors should also investigate and discuss the relationship between seismic amplitude and energy and how this would impact their analysis.

Significance and fit values The authors argue the significance of their trends based on

goodness of fit statistics such as R2. There are at least a few instance (especially when claiming no significant relation), where a non-parametric statistic such as Kendall's tau would be more appropriate.

Please note that all reviewers' comments that I have not mentioned should also be treated with due care.

2.30 These authors. . .

3.15 French Alps (French with capital letter)

5.3 . . .dependent on. . .

9.19 . . .for blocks for which. . .

9.25 . . .the uncertainties associated with determining the amplitude at the source are lower than those associated with seismic energy.

—————————————————

---

## Author Comment (AC1) · 22 Feb 2017

(We invite the readers to look at the version of this response uploaded as supplement for a better readability)

Response to Referee 1 (Major Points) :

Dear Maxime Farin,

We are very grateful for the reviews and comments you provided on our paper entitled "Single-block rockfall dynamics inferred from seismic signal analysis." We provided below answers to each of the major points raised. Several comments are repeated in the review. We responded to these repeated comments the first time they appear in the review and then refer to these answers when needed. The comments on specific

lines will be address in our final response. The answer to a comment is given after repeating the comment.

The authors.

Review by M. Farin

General comments :

MF: The paper is globally clear to read and the successive sections follow naturally each others. I fidnd personally that it is interesting to have new data of seismic signals generated by block impacts and be able to evaluate the dynamics of the block in parallel in order to better understand the link between the two on the field. The authors took care to evaluate the dynamics of the block with a good precision, with an uncertainty less than 1 m sˆ-1 for block speeds varying from 6 m sˆ-1 to 17 m sˆ-1. When we compare seismic parameters to dynamic parameters, it is important to evaluate the absolute seismic parameters at the source because they strongly depend on the distance between the source and the instrument and on the frequency. Care has also been taken in evaluating absolute seismic parameters in this paper. Therefore I think the presented data are of good quality. However, I think that the paper needs a major revision before being considered for publication because it contains major confusions and misinterpretations of the data. My main concern in is the fact that the authors say several times in the paper that they show a scaling (or proportionality) between the seismic amplitude and the momentum of the block while they are showing a linear relationship. There is a important confusion here because a scaling (or proportionality) is a relation $Y = aX$ while a linear relationship (as showed in this paper) is $Y = aX+b$, with b a nonzero constant. This has a different implication for the interpretation of the data. The paper should be rewritten with this point in mind. This confusion is particularly problematic when the authors are comparing the parameter $mVz^{(13/5)}$ derived by Farin et al. (2015) to the radiated seismic energy Es. They are testing a law $Es = a\,mVz^{(13/5)} + b$ and claim that the fit of this law with their data is better than it was

in the paper of Farin et al. (2015). However, the analytical scaling law established in Farin et al. (2015) and tested with their rockfall experiments was Es = a mVzˆ(13/5) (with b=0): this is a different law. In the present paper, the parameter b is not 0 and it is several orders of magnitude larger than the parameter a. The fit Es = a mVzˆ(13/5) (with b=0) should be tested instead. Moreover, since the parameter b does not exist in the analytical model, I do not know if this parameter has a physical meaning, even though it has the dimension of an energy. Also, an analytical expression of the proportionality coefiňĄcient a is given in Farin et al. (2015). The exact law and empirical law (with the exact and empirical value of a) could be compared to the seismic energy Es.

AC: Our first intention was to process the data for single rockfalls and seek for the best relationships as it was done in other studies on large landslides or rockfalls (e.g. Deparis et al., 2008, Hibert et al., 2017). In those studies, the best correlations were found using linear relationships, which naturally led us to use the same approach for this study. We agree, in the light of the comments made by the referees and the editor, that proportionality laws have to be tested too, and the confusion between linear and proportional relationships lifted.

To address this comment we computed proportional laws for each pair of quantities chosen. We modified table 2 to show these results. The new Table 2 is reproduced below. For the sake of clarity, we also decided to remove the coefficients computed in the logarithm space, as we discuss and use only the relationships computed in the linear space in the rest of the paper. We will also modify figure 4 to show the data in the linear space, and add the regression lines associated with proportional relationships. Table 1 : New table 2 – Coefficients of the regression lines for proportional and linear relationships.

As shown by this new table, the regression of our data by proportional laws yields slightly worst fits (lower R2 values), but with $\alpha$ coefficients very close to the one returned by linear regression. The coefficients $\beta$ in the linear regressions are close to zero (even if order of magnitude larger than coefficients $\alpha$). This might explain why the

coefficient $\alpha$ and R2 returned by the proportional relationships are very close to the one observed for the linear ones. The slightly better fit achieved by linear regressions might come from the accommodation of the uncertainties on the values of the tested parameters, which are inherent to the processing of real data.

The paper will be modified by taking into account these new results, however this will not impact the main conclusions of our work, which are: (i) Linear/proportional relationships exist between the maximum amplitude and the momentum, and between the seismic, the kinetic and the potential energies, and (ii) we can retrieve rockfalls properties directly from the seismic signals generated at impacts.

MF: - An interesting question when we study the seismic signal generated by rockfall is to establish their energy budget, i.e. determine the amount of kinetic energy or potential energy lost that is radiated in the form of elastic waves. In other words, I think the authors should compute the value of the ratios Es/Ek and Es/Ep (or maybe also Es/(Ek+Ep)). These ratios should be less than 1 and the rest of the kinetic and/or potential energy lost is dissipated in plastic deformation (irreversible deformation) of the ground or in viscoelastic processes (heat). These ratios can then be compared with that computed for larger rockfalls in the crater of the Piton de la Fournaise, La Reunion Island (Hibert et al. 2012) or with that obtained in other studies (e.g. Deparis et al. 2008). Thus we could see if the energy budget for one single impactor is different than for a rockfall constituted of several blocks. These ratios are proportionality relations between seismic and dynamic parameters.

AC: Those ratios are directly given by the relationships we found (see table above). We will add a comment in the discussion on these values, which are slightly lower than the one computed at Piton de la Fournaise or Soufrière Hills volcano (10-6 vs. 10-5 – 10-3). We suspect that the nature of the substrate (black-marls, i.e. soft sediments) can be the cause of these lower ratios.

MF: - In a nutshell, I think that proportionality relationships Y=aX between seismic and

dynamic parameters would have much more interesting implications for interpretations of the seismic signals generated by rockfalls than linear relationships Y=aX+b. Besides, no confusion should be made between the two kinds of relationship. A linear relationship may better fit the data of this paper than a proportionality law X = a Y but in this case, both fits (X = aY+b and X = aY) should be shown and a physical interpretation of parameter b should be given.

AC: see comment above.

MF: - An other problem I see is when the authors want to retrieve the mass and the speed of the blocks from the seismic signal. Two seismic variables are used: the absolute seismic amplitude and the radiated seismic energy. However, I do not think these two variables are independent of each others. I would not be surprised if the radiated seismic energy is proportional to the squared absolute amplitude. In this case, the mass and the speed could be expressed as functions of the radiated seismic energy alone. The problem is that I don't think it is possible to retrieve two independent dynamic parameters from only one seismic variable.

AC: We do not correlate the absolute seismic amplitude to the momentum but to the maximum of the amplitude envelope. This is an important distinction as the peak amplitude might not be correlated to the seismic energy (integral of the envelope). For example, a long –duration seismic signal with no clear peak amplitude might have the same seismic energy as an impulsive, high–amplitude, short–duration seismic signal. As shown by the figure below with our data, these quantities are not dependant in our case.

MF: An advantage of the present study compared with the previous ones (e.g. Farin et al. (2015)) is that the authors have access to higher frequencies up to 500 Hz, with respect to 50 Hz before. Therefore, they potentially have access to all the frequencies emitted during the impacts, contrary to the previous study. Thus an interesting seismic parameter to evaluate would be the mean frequency of the seismic signal.

the analytical model of impact of Hertz shows that the mean frequency is inversely proportional to the mass m of the block. It would be interesting to test this scaling. The mean frequency of the signal is independent of the radiated seismic energy so if empirical scaling laws are established between these two parameters and the mass and the speed of the block, the laws can be inverted to retrieve the masses and the speeds. Farin et al. (2015) established two analytical scaling laws relating the mass and the speed of the block to the radiated seismic energy and the mean frequency of the signal, i.e. equations (29) and (30) of their paper. I would be curious to see if these equations can provide reasonable values of the masses m and the speeds Vz of the blocks with the present experiments.

AC: Regarding an approach based on the frequency content, there are two limitations. The first one is that the seismometer located down-slope has a Nyquist frequency of 50 Hz. Hence, we had to restrict our study to the 1-50 Hz frequency band, as most of the times we need this station to compute the attenuation parameters and thus the amplitude and the energy at the source. Second, because we are lacking a good propagation model, we cannot reconstruct the Green's function of the medium between the location of each impact and the stations. Without these Green's functions, it is impossible to extract the frequency content of the source. This prevents any analysis of the frequency content of the seismic signal of each impact, as we cannot decipherer source effect from propagation effect. As clearly shown by Figure 2b, the major control on the frequency content of seismic signal recorded at each impact is related to its distance to the station. Therefore it makes no sense to compute the average frequency, as it is predominantly controlled by the medium and not the source.

This underlies that an implementation of a frequency-based approach for the quantification of rockfall properties from the seismic signal they generate would be difficult in an operational context. The new approach we propose in this study does not require a thorough characterization of the medium, and we show that we can determine rockfalls properties simply from the seismic signal temporal features. We will add a paragraph

**ESurfD**
* * *
Interactive
comment

in the discussion about this point.

MF: - Maybe the absolute seismic amplitude and the radiated seismic energy are independent of each others. In that case it should be shown somewhere. Besides, if the mean frequency of the signal is not inversely proportional to the mass of the block, it would be interesting to show it. That would mean that Hertz's model does not apply on the field.

AC: see comment above

Please also note the supplement to this comment:
http://www.earth-surf-dynam-discuss.net/esurf-2016-64/esurf-2016-64-AC1-supplement.pdf

$R^2$=0.27

**Fig. 1.** Squared maximum amplitude A0 as a function of the energy of the seismic signal generated at each impact

| Parameters $(X, Y)$ | Proportional | | | linear | | |
|---|---|---|---|---|---|---|
| | $\alpha$ | $\beta$ | $R^2$ | $\alpha$ | $\beta$ | $R^2$ |
| $A0_{max} = \alpha p + \beta$ | $2.35\,10^{-9}$ | 0 | 0.63 | $2.26\,10^{-9}$ | $2.50\,10^{-7}$ | 0.64 |
| $E_s = \alpha E_p + \beta$ | $-4.40\,10^{-6}$ | 0 | 0.61 | $-5.04\,10^{-6}$ | $-0.01$ | 0.61 |
| $E_s = \alpha E_k + \beta$ | $2.59\,10^{-6}$ | 0 | 0.59 | $3.09\,10^{-6}$ | $-0.01$ | 0.64 |
| $E_s = \alpha m + \beta$ | $1.48\,10^{-4}$ | 0 | 0.23 | $2.85\,10^{-4}$ | $-0.03$ | 0.31 |
| $E_s = \alpha m V_z^{13/5} + \beta$ | $4.86\,10^{-7}$ | 0 | 0.62 | $5.85\,10^{-7}$ | $-0.01$ | 0.63 |
| $E_s = \alpha m V_z^{0.5} + \beta$ | $5.24\,10^{-5}$ | 0 | 0.33 | $1.07\,10^{-4}$ | $-0.04$ | 0.47 |

**Fig. 2.** New table 2 – Coefficients of the regression lines for proportional and linear relationships.

---

## Author Response (AR1)

**Final Response**

Dear Editor and Referees,

We are very grateful for the reviews and comments you provided on our paper entitled "Single-block rockfall dynamics inferred from seismic signal analysis." Below we respond to the comments and suggestions provided.

We include the response made to the major points discussed by Referee 1 as in this letter we refer to answers we provided on several of the issues he raised.

The answer to a comment is given after repeating the comment and colored in **blue**. When changes have been made in the manuscript to take into account a suggestion we indicate the corresponding lines and page in the marked-up manuscript. The marked-up manuscript version showing the changes made is provided after our responses.

We hope that the revised manuscript is now suitable for a publication in *E-Surf.*

The authors.
* * *
**JM Turowski (Editor) :**

Dear authors, We have now received two reviews, and, since the discussion closes in a few days, I give a short summary of what I find important. While both reviewers are generally supportive of the manuscript, they both raise some criticisms that need to be addressed before publication. I add some comments of my own reading to that.

Presentation and language Reviewer #2 criticizes presentation and language and asks for rewriting with a focus on clarity. I agree that there are fairly frequent odd formulations and unclear writing.

We tried to improve the clarity and the overall language in the whole manuscript.

Scaling, linearity and the fits Reviewer #1 raises concerns about the terminology used in the paper. I do not agree with his definitions; in my understanding, two variables A and B scale with each other, if they have a positive monotonic relationship, without the need of specifying a function. That is, if A increases, B increases also. Two variables are proportional if their ratio is a constant. And they are linearly related if A=mB+b, where m and b are constants. However, I agree with reviewer #1 that in the manuscript, the terminology is not used in a common way. For example, I would not say that two variables scale linearly, but rather that they have a linear relationship of linear dependence. That said, there is something funny about the plots in Fig. 4 and the way the relationships are discussed. The plots in Fig. 4 are all log-log. In this representation, proportionality would result in a straight line with a gradient of one. A linear relationship would result in a curved line. A closer look reveals that the depicted fit lines actually have gradients that deviate from one. They show power law relationships. The fit values given in Table 2 indicate exponents of up to 2. This may change the entire results, discussion and outcomes of the paper. Here is at least a major problem with the communication, if not with the use of the fits and the statistical relationships. These need to be carefully resolved and communicated.

Following this remarks and the suggestions of Referee 1 we reorganized the paper and changed significantly the results section of the paper **(p.8-p.11)**. We clarified our approach and the terminology used as suggested.

Regarding the power-law assumption : There are indeed $\alpha$ coefficients close to 2 when looking at the regression laws found in the log-log space, but those coefficients are obtained for pair of parameters that we think are not strongly correlated ("Seismic energy" and "mass" for example). Moreover, the uncertainty of those coefficients is large. For example, with the pair of parameters "seismic energy" and "kinetic energy", $\alpha$=1.38 but with 95% coefficients bounds of +/- 0.4. Moreover, all the other studies (e.g. Deparis et al., 2008; Hibert et al., 2011, 2017; Yamada et al., 2012; Levy et al., 2015) that sought correlations between the seismic features and the event properties found linear relationships. Therefore we decided to remove the results obtained in the log-log space and we included a short-paragraph in the "results" section to explain why we choose to fit the data with linear relationships **(p.10 l.26-29)**.

Relation to theory Reviewer #1 comments that the relations to his theory are partially incorrect and not well described. In light of the issue raised in the preceding point, I ask the authors to clearly present the used theory in the paper, identify appropriate hypotheses that can be tested with the data (both a theory-derived hypothesis and a null hypothesis), and to discuss how the outcomes of their experiments relate to these.

We agree and added a paragraph **(p.8 l.13-18 and p.10 l.1-8)** explaining the assumptions tested.

Energy budget I also like reviewer #1's suggestion of the energy budget and ask the authors to provide appropriate calculations and a discussion on this.

We agree and added a paragraph in the discussion **(p.13 l.20-32)** (also see response to Referee 1 comment).

Relationship between seismic energy and amplitude The authors should also investigate and discuss the relationship between seismic amplitude and energy and how this would impact their analysis.

We computed a correlation coefficient between these two parameters (see comment below) and included it in Table 1 to show that they are not correlated. (Also see response to the comment of Referee 1 on this issue).

Significance and fit values. The authors argue the significance of their trends based on C2 goodness of fit statistics such as R2. There are at least a few instance (especially when claiming no significant relation), where a non-parametric statistic such as Kendall's tau would be more appropriate.

We agree and we modified the "result section" accordingly. We chose the Spearman's rho non-parametric statistic measure as we assume that our data should scale following a monotonic law. **(p.10 l.7-21)**

Please note that all reviewers' comments that I have not mentioned should also be treated with due care.

2.30 These authors. . .

3.15 French Alps (French with capital letter)

5.3 . . .dependent on. . .

9.19 . . .for blocks for which. . .

9.25 . . .the uncertainties associated with determining the amplitude at the source are lower than those associated with seismic energy.

**Referee 1 :**

**Review by M. Farin**

**General comments :**

The paper is globally clear to read and the successive sections follow naturally each others. I find personally that it is interesting to have new data of seismic signals generated by block impacts and be able to evaluate the dynamics of the block in parallel in order to better understand the link between the two on the field. The authors took care to evaluate the dynamics of the block with a good precision, with an uncertainty less than 1 m s^-1 for block speeds varying from 6 m s^-1 to 17 m s^-1. When we compare seismic parameters to dynamic parameters, it is important to evaluate the absolute seismic parameters at the source because they strongly depend on the distance between the source and the instrument and on the frequency. Care has also been taken in evaluating absolute seismic parameters in this paper. Therefore I think the presented data are of good quality. However, I think that the paper needs a major revision before being considered for publication because it contains major confusions and misinterpretations of the data.

- My main concern in is the fact that the authors say several times in the paper that they show a scaling (or proportionality) between the seismic amplitude and the momentum of the block while they are showing a linear relationship. There is a important confusion here because a scaling (or proportionality) is a relation Y = a X while a linear relationship (as showed in this paper) is Y = aX+b, with b a nonzero constant. This has a different implication for the interpretation of the data. The paper should be rewritten with this point in mind. This confusion is particularly problematic when the authors are comparing the parameter mVz^(13/5) derived by Farin et al. (2015) to the radiated seismic energy Es.  They are testing a law Es = a mVz^(13/5) + b and claim that the fit of this law with their data is better than it was in the paper of Farin et al. (2015). However, the analytical scaling law established in Farin et al. (2015) and tested with their rockfall experiments was Es = a mVz^(13/5) (with b=0): this is a different law. In the present paper, the parameter b is not 0 and it is several orders of magnitude larger than the parameter a. The fit Es = a mVz^(13/5) (with b=0) should be tested instead. Moreover, since the parameter b does not exist in the analytical model, I do not know if this parameter has a physical meaning, even though it has the dimension of an energy. Also, an analytical expression of the proportionality coefficient a is given in Farin et al. (2015). The exact law and empirical law (with the exact and empirical value of a) could be compared to the seismic energy Es.

Our first intention was to process the data for single rockfalls and seek for the best relationships as it was done in other studies on large landslides or rockfalls (e.g. *Deparis et al.*, 2008, *Hibert et al.*, 2017). In those studies, the best correlations were found using linear relationships, which naturally led us to use the same approach for this study. We agree, in the light of the comments made by the referees and the editor, that proportionality laws have to be tested too, and the confusion between linear and proportional relationships lifted.

To address this comment we computed proportional laws for each pair of quantities chosen. We modified table 2 to show these results. The new Table 2 is reproduced below and in the revised

version of the manuscript is relabelled as Table 1. For the sake of clarity, we also decided to remove the coefficients computed in the logarithm space, as we discuss and use only the relationships computed in the linear space in the rest of the paper. We will also modify figure 4 to show the data in the linear space, and add the regression lines associated with proportional relationships.

Table 1 : New table 1 : Spearman correlation coefficients, coefficients of the regression lines for proportional and linear relationships and corresponding coefficient of determination

| Parameters $(X,Y)$ | Spearman correlation | | Proportional | | | linear | | |
|---|---|---|---|---|---|---|---|---|
| | $\rho$ | $p-values$ | $\alpha$ | $\beta$ | $R^2$ | $\alpha$ | $\beta$ | $R^2$ |
| $A0_{max} = \alpha|p| + \beta$ | 0.67 | $1.12\,10^{-7}$ | $2.35\,10^{-9}$ | 0 | 0.63 | $2.26\,10^{-9}$ | $2.50\,10^{-7}$ | 0.64 |
| $E_s = \alpha E_p + \beta$ | 0.68 | $6.75\,10^{-6}$ | $4.40\,10^{-6}$ | 0 | 0.61 | $5.04\,10^{-6}$ | -0.01 | 0.61 |
| $E_s = \alpha E_k + \beta$ | 0.70 | $3.01\,10^{-6}$ | $2.59\,10^{-6}$ | 0 | 0.59 | $3.09\,10^{-6}$ | -0.01 | 0.64 |
| $E_s = \alpha m + \beta$ | 0.51 | $1.3\,10^{-3}$ | $1.48\,10^{-4}$ | 0 | 0.23 | $2.85\,10^{-4}$ | -0.03 | 0.31 |
| $E_s = \alpha m V_z^{13/5} + \beta$ | 0.69 | $4.16\,10^{-6}$ | $4.86\,10^{-7}$ | 0 | 0.62 | $5.85\,10^{-7}$ | -0.01 | 0.63 |
| $E_s = \alpha m V_z^{0.5} + \beta$ | 0.62 | $7.63\,10^{-5}$ | $5.24\,10^{-5}$ | 0 | 0.33 | $1.07\,10^{-4}$ | -0.04 | 0.47 |
| $A0_{max} = \alpha E_s + \beta$ | 0.44 | $8.2\,10^{-3}$ | - | - | - | - | - | - |

As shown by this new table, the regression of our data by proportional laws yields slightly worst fits (lower $R^2$ values), but with $\alpha$ coefficients very close to the one returned by linear regression. The coefficients $\beta$ in the linear regressions are close to zero (even if order of magnitude larger than coefficients $\alpha$). This might explain why the coefficient $\alpha$ and $R^2$ returned by the proportional relationships are very close to the one observed for the linear ones. The slightly better fit achieved by linear regressions might come from the accommodation of the uncertainties on the values of the tested parameters, which are inherent to the processing of real data. We added a comment on this point in the revised manuscript **(p.11 l.4-6).**

The paper will be modified by taking into account these new results, however this will not impact the main conclusions of our work, which are: (i) Linear/proportional relationships exist between the maximum amplitude and the momentum, and between the seismic, the kinetic and the potential energies, and (ii) we can retrieve rockfalls properties directly from the seismic signals generated at impacts.

- An interesting question when we study the seismic signal generated by rockfall is to establish their energy budget, i.e. determine the amount of kinetic energy or potential energy lost that is radiated in the form of elastic waves. In other words, I think the authors should compute the value of the ratios Es/Ek and Es/Ep (or maybe also Es/(Ek+Ep)). These ratios should be less than 1 and the rest of the kinetic and/or potential energy lost is dissipated in plastic deformation (irreversible deformation) of the ground or in viscoelastic processes (heat). These ratios can then be compared with that computed for larger rockfalls in the crater of the Piton de la Fournaise, La Reunion Island (Hibert et al. 2012) or with that obtained in other studies (e.g. Deparis et al. 2008). Thus we could see if the energy budget for one single impactor is different than for a rockfall constituted of several blocks. These ratios are proportionality relations between seismic and dynamic parameters.

Those ratios are directly given by the relationships we found (see table above). We will add a comment in the discussion on these values, which are slightly lower than the one computed at Piton de la Fournaise or Soufrière Hills volcano ($10^{-6}$ vs. $10^{-5} - 10^{-3}$). We suspect that the nature of the substrate (black-marls, i.e. soft sediments) can be the cause of these lower ratios **(p.13 l.20-32).**

- In a nutshell, I think that proportionality relationships Y=aX between seismic and dynamic parameters would have much more interesting implications for interpretations of the seismic signals generated by rockfalls than linear relationships Y=aX+b. Besides, no confusion should be made between the two kinds of relationship. A linear relationship may better fit the data of this paper than

a proportionality law X = a Y but in this case, both fits (X = aY+b and X = aY) should be shown and a physical interpretation of parameter b should be given.

(see comment above).

- An other problem I see is when the authors want to retrieve the mass and the speed of the blocks from the seismic signal. Two seismic variables are used: the absolute seismic amplitude and the radiated seismic energy. However, I do not think these two variables are independent of each others. I would not be surprised if the radiated seismic energy is proportional to the squared absolute amplitude. In this case, the mass and the speed could be expressed as functions of the radiated seismic energy alone. The problem is that I don't think it is possible to retrieve two independent dynamic parameters from only one seismic variable.

We do not correlate the absolute seismic amplitude to the momentum but to the **maximum** of the amplitude envelope. This is an important distinction as the **peak** amplitude might not be correlated to the seismic energy (integral of the envelope). For example, a long –duration seismic signal with no clear peak amplitude might have the same seismic energy as an impulsive, high–amplitude, short–duration seismic signal. As shown by the computed Spearman's correlation coefficient (Table) and the figure below with our data, these quantities are not dependant in our case.

[Figure]

**Figure 1 :** Squared maximum amplitude A0 as a function of the energy of the seismic signal generated at each impact

An advantage of the present study compared with the previous ones (e.g. Farin et al. (2015)) is that the authors have access to higher frequencies up to 500 Hz, with respect to 50 Hz before. Therefore, they potentially have access to all the frequencies emitted during the impacts, contrary to the previous study. Thus an interesting seismic parameter to evaluate would be the mean frequency of the seismic signal. the analytical model of impact of Hertz shows that the mean frequency is inversely proportional to the mass m of the block. It would be interesting to test this scaling. The mean frequency of the signal is independent of the radiated seismic energy so if empirical scaling laws are established between these two parameters and the mass and the speed of the block, the laws can be inverted to retrieve the masses and the speeds. Farin et al. (2015) established two analytical scaling laws relating the mass and the speed of the block to the radiated seismic energy and the mean frequency of the signal, i.e. equations (29) and (30) of their paper. I would be curious to see if these equations can provide reasonable values of the masses m and the speeds Vz of the blocks with the present experiments.

Regarding an approach based on the frequency content, there are two limitations. The first one is that the seismometer located down-slope has a Nyquist frequency of 50 Hz. Hence, we had to restrict our study to the 1-50 Hz frequency band, as most of the times we need this station to compute the attenuation parameters and thus the amplitude and the energy at the source. Second,

because we are lacking a good propagation model, we cannot reconstruct the Green's function of the medium between the location of each impact and the stations. Without these Green's functions, it is impossible to extract the frequency content of the source. This prevents any analysis of the frequency content of the seismic signal of each impact, as we cannot decipherer the source effects from the propagation effects. As clearly shown by Figure 2b, the major control on the frequency content of the seismic signal recorded at each impact is related to its distance to the station. Therefore it makes no sense to compute the average frequency, as it is predominantly controlled by the medium and not the source.

This underlies that an implementation of a frequency-based approach for the quantification of rockfall properties from the seismic signal they generate would be difficult in an operational context. The new approach we propose in this study does not require a thorough characterization of the medium, and we show that we can determine rockfalls properties simply from the seismic signal temporal features.

- Maybe the absolute seismic amplitude and the radiated seismic energy are independent of each others. In that case it should be shown somewhere. Besides, if the mean frequency of the signal is not inversely proportional to the mass of the block, it would be interesting to show it. That would mean that Hertz's model does not apply on the field.

(See comment above)

**Comments on specific lines in the paper:**

- The abstract needs a context sentence.

We added a context sentence **(p.1 l.2-3)**

- page 1 line 8, line 10…, ' the energy of the corresponding part of the seismic signal ' , 'the energy of the seismic radiation ',… try to always call this energy is the same way all along the paper, for example ' the radiated seismic energy ' because it is sometimes difficult to understand to what energy you are referring to.

We agree and made the proposed modification.

- page 1 line 8: ' Our results suggest that the amplitude of the seismic signal scales with the momentum of the block at the impact. '. No, be careful thorough in the paper: a scaling is a proportionality, not a linear relationship. This is important.

Revised.

- page 1, line 12: ' the masses and the velocities ' or ' the mass and the velocity '

Revised

- page 2, line 19: precise that this is true in the frequency range 3 Hz to 10 Hz

Precision added.

- page 2, line 20: 'The authors also demonstrated that the maximum amplitude of the seismic signal, corrected from propagation effects, scales with the bulk momentum ': That was also a linear relationship, not a scaling (i.e. proportionality).

Revised

- page 2, lines 27-32: this paragraph needs rewriting: line 30: ' The impulse imparted to the solid Earth by a bouncing particle within a granular flow will be proportional to the kinematics of the particle, and the amplitude of the seismic wave will be proportional to the magnitude of the impulse '. This sentence is not very clear, it particular 'proportional to the kinematics of the particle ' does not mean anything:

This suggests that we do not know yet which kinematic parameters control the impulse that generates the seismic waves. It could be the velocity, but also the momentum, the kinetic energy, etc. We modified slightly this sentence to make this point clearer. **(p.3 l.4)**

- page 2, l. 34: not exactly true: the mass and the speed of an impactor can be related to the radiated elastic energy and the mean frequency of the signal. Do not write ' at a given frequency ', it could be misinterpreted.

Agreed and revised.

- page 3, l.1: precise here what is the relation you are referring to: $Es = a mv^{(13/5)}$.

Here we are not only referring to this relation but to the several tested in Farin et al., (2015). This relation, as well as others, is discussed later in the paper and we do not think this equation needs to be already explicitly mentioned in the introduction.

- page 3, l.4: It is very strange for me why you say that having frequency < 50 Hz is a limitation (which is true) but then you are filtering your signals below 50 Hz in the following (p 6, l 30). Why don't you take advantage of having frequencies higher than 50 Hz in order to improve the estimate of the masses and speeds of the blocks with respect to the previous study? Moreover, you know the mass and the speed of the blocks so you can evaluate a theoretical mean frequency of the signal generated by an impact using Hertz theory of impact and then compare with the measured mean frequency in your data. You would know if your seismic stations are sensitive to the whole frequency spectrum emitted during the impact or if you are loosing energy in the highest frequency (if the measured mean frequency is smaller than the theoretical mean frequency). This would help you to interpret the difference between the measured radiated seismic energy Es and the parameter a $mV^{(13/5)}$: normally the measured Es should be smaller than a $mV^{(13/5)}$ at high frequency if you are not sensitive to the highest frequencies. Finally, if you don't obtain any satisfying results using the mean frequency, it would mean that the mean frequency is not a reliable enough parameter to use to extract information from the seismic signal generated by impacts on the field: this is an interesting result.

(See general comments above)

- page 3, l. 4, you should rather say ' a great part of the energy liberated at the impact is at high frequencies (> 50 Hz) …'. Another important limitation we had was that there was no synchronization between the seismic signal and the movies…

We rewrote this sentence. **(p.3 l.14-16)**

- page 3, l.6-11: You should better highlight what is new in your study with respect to the previous study: you use several seismic stations that can record higher frequencies, up to 500 Hz and a better identification of the seismic signals associated with each impact of the blocks.

We completed this sentence as suggested.

- page 3, l.25: Is the torrent producing a lot of seismic noise?

No, the torrent was almost dry at the time of the experiment.

- page 4, l.6: Define clearly what are the potential energy lost and the kinetic energy as a function of the mass and the speed of the block, and show these relations on the axis of Figure 4. Also: is the speed of impact Vz vertical or inclined with respect to the slope/vertical? Can you observe an effect of the angle of impact with respect to the normal to the slope on the radiated seismic energy? Are more inclined impacts less seismically efficient (lower Es/Ep) than more normal impacts? This might potentially explain part of the discrepancy.

Vz is the vertical velocity. We added the equations used to compute the kinetic and the potential energies **(Eq. 1 and Eq. 2)**. We do not think showing these equations on the axes of figure 4 is useful.

We do not observe an effect of the angle of impact on the energies ratio but this might not be significative as the uncertainties on the computation of the seismic energy are large.

- page 4, l. 11: Write here the range of values the speed of impacts Vz can take because we don't know if 1 m s^-1 is a large and small uncertainty.

Precision added.

- Figure 1c: not clear what the colored points are referring to. The text is the figure is too small, especially along the torrent.

Precision added in the caption. We increased the font size of the text.

- page 5, l. 4, the Figure 2b also shows the attenuation of small frequencies. Rephrase the sentence.

We do not understand this comment. Figure 2b predominantly shows the attenuation of high frequency waves. Lower frequencies are visible at each impact, hence this figure does not illustrate any attenuation of the low-frequency seismic waves.

- page 5: It is not clear how you obtain the equation (2) and what the index ij are representing for B. You should directly say that B depends on frequency and show B as a function of the frequency, on a Figure for example (maybe in Appendix), so that we can know what is the quality factor Q. If you assume that B does not vary with frequency then give the value of B (or a range of values).

*Bij* is the average values of the anelastic attenuation coefficient *B* for an impact recorded on station *i* and station *j,* as explained on lines **5-7 p6**. We do not think giving the corresponding value of Q is relevant, as *B* is an *apparent* anelastic attenuation parameter, which may not reflect the true attenuation coefficient of the medium.

- page 5, l. 23: Can you measure the wave speed in this specific site with your present seismic data by measuring the difference of time travel after an impact between several seismic stations?

This should be possible but would require significant processing and is not the main focus of this paper.

- page 6, l. 30: What a pity not to use the high frequencies > 50 Hz. There may a lot of interesting information in it.

(see general comment)

- Figure 4:

- Do you observe a correlation between radiated seismic energy Es and the squared momentum |p|^2 ?

(see general comment)

- See my first comment about the law X = a Y and the energy ratios.

- See my first comment: the laws established and tested in Farin et al. (2015) are Es = a mVz^(13/5) and Es = a mVz^(0.5), not the ones you are showing.

- The caption of the figure can be simplified: ' Decimal logarithm of the seismic energy Es of the seismic signal generated at the impact as a function of (b) Ek, (c) Mass, (d) Ep, (e) mVz^(0.5) and (f) mVz^(13/5). '

Caption revised.

- page 9, l. 3-9: Rewrite this paragraph in accordance with my first comment.- page 9, eq. (6): I am not sure that the absolute amplitude and the radiated elastic energy are independent variables. Also write explicitly the equation for the speed Vi as a function of the signal parameters.

We added the equation for the speed Vi (Eq. 8).

- Table 1, Fig 5: Represent the results of the inversion more uniformly: a figure with two plots showing (a) mi as a function of mr and (b) Vi as a function of Vr, with error bars and the line Y=X would be much clearer than a table and a histogram that mean to represent the same thing for m and Vz.

We think that a table is more appropriate, as it includes the number of impacts used to compute the inferred mass. As commented in the text, the number of impacts used reduces the uncertainty on the determination of the mass. This is important to show. The histogram, is, in our opinion, easier to interpret for the readers.

- page 11, l.4: ' linear scaling ' => linear relationships.

Revised.

- page 11, l. 7: ' the seismic radiation released at each impact scales linearly with the potential energy lost ': no.

We meant here that the seismic energy radiated is correlated with the potential energy, which is true according to our result. We modify this sentence to make this point clear.

- page 11, paragraph 2:

- You did not verify this scaling law either.

- ' In our study the instruments we deployed permitted to record most of the energy generated at impacts. This underlines the importance of choosing adequate seismometers, capable of recording the whole seismic energy generated at the impacts,  for future studies. ': Yes but you did not take advantage of this because you filtered the signals below 50 Hz while energy is clearly visible at more than 200 Hz on Figure 2a… Therefore you can not use this sentence to explain why you observed better correlations.

We agree and modified this paragraph.

- page 11, last line: ' We show that the maximum amplitude of the seismic signal generated by the impact of a single particle is proportional to its momentum. ' This is false.

According to our results, this is true.

- page 12, first line: 'The source of the seismic signal generated at this given time might therefore be the sum of the impulses imparted by the particles to the ground. ' I do not think we can say that because the signals emitted by two particles impacting the ground at roughly the same position and time can destruct or add themselves, depending of their phase. The energies of each impacts may be added, however (see the paper of Tsai et al. 2012 on the seismic noise of river: ' A physical model for seismic noise generation from sediment transport in rivers ', GRL (2012).). Moreover, in the granular flow experiments we did with Anne Mangeney during my PhD (cf. Farin, M. (2015), Étude expérimentale de la dynamique et de l'émission sismique des instabilités gravitaires. IPGP, France.), we showed that the scaling law that relate the radiated seismic energy to the mass for a single impactor is not the same as for a granular flow of multiple particles of the same size. The relationship between the radiated seismic energy and the mass and the speed of the particles in a granular flows is much more complex than for one impact because all the particles are interacting with each others and each of them move in a random direction with respect to its neighbors (in an agitated flow) and each of them has a different fluctuating speed (instantaneous speed - mean speed of the flow). Therefore, the seismic amplitude generated by a granular flow does not simply scale (nor has a linear relationship) with the momentum of one particle in the flow. As you say in the last sentence, numerical models (DEM or statistical models like kinetic theories of granular gas) can help us better understand the complex link between particle/flow dynamics and seismic signal in granular flows.

We agree with those insightful remarks and moderated the last paragraph of the discussion.
* * *
**Referee 2 :**

**Review by F. Panzera**

The manuscript "Single-block rockfall dynamics inferred from seismic signal analysis" by Hibert et al. is interesting and contains innovative information about seismic radiation due to rockfall. Below some comments that I hope help the authors in improving the manuscript.

Although I am not an English mother-tongue, at times I found some sentences difficult to follow. In my opinion, the authors should improve the English language.

We tried to identify and improve the sentences that were difficult to understand.

In the Introduction, few lines should be added on the importance of rockfalls characterization, through seismic method (but not only). See for instance:

Burjanek J., Moore J.R., Yugsi-Molina F.X., Fah D. (2012) Instrumental evidence of normal mode rock slope vibration. Geophys. J. Int., 188, 559–569.

F. Panzera, S. D'Amico, A. LotC1 teri, P. Galea, G. Lombardo (2012) Seismic site response of unstable steep slope using noise measurements: the case study of Xemxija bay area, Malta. Nat. Hazard Earth Sci. Syst., 12, 3421–3431 doi: 10.5194/nhess-12-3421-2012

P. Galea, S. D'Amico, D. Farrugia (2014) Dynamic characteristics of an active coastal spreading area using ambient noise measurements – (Anchor Bay, Malta). Geophys. J. Int., 199, 1166–1175 doi: 10.1093/gji/ggu318.

These studies are not focused on the relationships between the dynamics of rockfalls and the seismic signal they generate but more on the use of the seismology to monitor the state of instable slopes or cliffs. However we agree that this other use of the seismology to mitigate risks associated with mass movements should be mentioned in the introduction, and we included these references and others [Amitrano et al., 2005; Levy et al. 2011]. **(p.2 l.1)**

In Figure 1a and b, what do the authors indicate with blue points? Which is the meaning of coloured points in Figure 1c?

As stated in the caption the blue points on figure 1a and b are the ground control points. We added the information on the meaning of the coloured points corresponding to the trajectory in the caption.

In Figure 1c, I understand that CMG1 is the broadband seismometer, but it is unclear which is the 3D short-period seismometer between K1, K2, K3 and K4. The authors should add a legend in map or some description in the figure caption.

We completed the figure caption.

The authors assume that seismic wavefield, generated by rockfalls, is composed mainly by surficial waves and consequently that the contribution of body waves is negligible. They must support this hypothesis through observations or by quoting references.

We provided references **(p.5 l.1-3)**

The authors assume that seismic wave velocity in black-marls is 300 m/s quoting as references Hibert et al. (2012) and Gance et al. (2012). Are the quoted studies performed in the same formations near Rioux Bourdoux? The sentence must be rewrite as follow: "The average velocity of surface waves in black-marls in the area of Rioux Bourdoux is approximately 300 m/s (Hibert et al., 2012; Gance et al., 2012)." or "The average velocity of surface waves in black-marls, considering information coming from literature, is approximately 300 m/s (Hibert et al., 2012; Gance et al., 2012)."

Yes the studies to which we refer to were done in the same formation (black-marls). We modified the sentence as suggested.

I suppose that the propagation depth is obtained by considering lampda=V/f. This assumption should by quoted in the text and the authors must specify why they chosen 20 Hz as central frequency for their computation.

We added this clarification in the text **(p.6 l.21)**

The authors used a linear regression to interpolated their data. Did they try to use a C2 power or logarithm law using a lin-lin graph? The R2 for each linear regression curve should be visible in the graphs of Figures 4.

We added the $R^2$ of each regression in Figure 4. Regarding other laws, see the response to the editor comment.

Probably in the case of x and y having uncertainties a Generalized Orthogonal Regression is need instead than standard least-squares.

Most of the studies we refer to in this manuscript used to quantify the good fit of their data with the standard least-squares estimator. For an easier comparison of our results with the ones already discussed in the literature we find it more convenient to use the same estimator.

The term "proportional" used in the manuscript is not correct, because the authors use a linear regression (y=ax+b) with a non-zero "b".

(See response to the general comment of referee 1)

[revised manuscript text omitted]

Coefficients of the regression lines

---

## Author Response (AR2)

**Final Response**

Dear Editor,

We are very grateful for the comments and suggested revisions you provided on our paper entitled "Single-block rockfall dynamics inferred from seismic signal analysis."

The answer to a comment is given below after repeating the comment, and colored in **blue**. When changes have been made in the manuscript to take into account a suggestion we indicate the corresponding lines and pages in the marked-up manuscript. The marked-up manuscript version showing the changes made is provided after our responses.

We hope that the revised manuscript is now suitable for a publication in *E-Surf.*

Best regards,

The authors.
* * *
**JM Turowski (Editor) :**

1.14/12.30 Here, the authors highlight the uncertainties. This point has not been treated in the discussion, but it would be good to have a paragraph on it.

Following this suggestion we expanded the discussion and included a paragraph on uncertainties **(p.12 l.9-17).**

1.15 The last sentence of the abstract is unspecific. What insights did you gain?

We clarified this statement

2.17 Ekström and Stark could be cited here as well.

In this section we speak about the relationships between high-frequency seismic signals and landslide properties. Ekström and Stark [2013] have worked on the inversion of low-frequency seismic signals, hence we do not think that it is relevant to refer to this study here.

2.35 this statement does not make sense, an impulse cannot be proportional to its dynamics (I have no idea what this means). Rewrite.

We mean here that the impulse, in a seismological source sense, might be proportional to the dynamic properties of the bouncing particle at each impact. We clarified this sentence **(p.2 l.35 – p.3 l.1-4)**.

3.30 How was time synchronization done? Did the camera have its own GPS transceiver or was this done manually?

The cameras had GPS transceiver.

4.7 reformulate more carefully. The pixel size essentially gives the error of the location. There will also be an error associated with the orthorectification procedure. Can you say something about this?

Unfortunately, we cannot determine precisely the error associated with the orthorectification. What we can estimate is the global average uncertainty on the position of the impacts from the difference of the positions determined from each camera.

Acknowledgements: it is customary to thank the reviewers.

This is an unfortunate oversight. We completed the acknowledgements **(p.14 l.7-8)**.

[revised manuscript text omitted]